# Explanations are a Means to an End: Decision Theoretic Explanation Evaluation

**Ziyang Guo** [1]   **Berk Ustun** [2]   **Jessica Hullman** [1]

## Abstract

Explanations of model behavior are commonly evaluated via proxy properties weakly tied to the purposes explanations serve in practice. We contribute a decision theoretic framework that treats explanations as information signals valued by the expected improvement they enable on a specified decision task. This approach yields three distinct estimands: (i) a theoretical benchmark that upper-bounds achievable performance by any agent with the explanation, (ii) a human-complementary value that quantifies the theoretically attainable value that is not already captured by a baseline human decision policy, and (iii) a behavioral value representing the causal effect of providing the explanation to human decision-makers. We instantiate these definitions in a practical validation workflow, and apply them to assess explanation potential and interpret behavioral effects in human–AI decision support and mechanistic interpretability.[1]

## 1. Introduction

Explanations have become a cornerstone in how people interact with machine learning models, from decision support [3] to regulatory compliance [16, 65] to mechanistic interpretability to drive model understanding and improvement [18, 51]. Validation of new methods, however, often disregards functional aspirations. Rather than asking *what an explanation is for*, work frequently targets abstract properties: objective internal criteria such as faithfulness [32], robustness to perturbations [2, 68], and the recovery of causal pathways [24, 49], or human-centered evidence of understanding, like users' self-reported appraisals of interpretability [40] or ability to predict a model's output [20].

Validation protocols used for explanation and interpretability techniques have long been criticized for relying heavily on authors' intuitions and anecdotal evidence [1, 41, 48, 53, 62]. Mounting empirical evidence also suggests limited effectiveness as decision support: a recent meta-analysis of 370 effects in studies of AI-assisted human decisions finds that providing an explanation did not lead to significantly improved decision quality [64].

In response, researchers have called for a more functional perspective that links explainability and interpretability to practice rather than pursuing understanding for its own sake [10, 52, 60]. One straightforward criterion is demonstrated improvement on a concrete task (e.g., pragmatic turns in mechanistic interpretability [52]). However, task choice and performance interpretation remain underspecified. Without a theoretical grounding for defining best-case use of an explanation and its *potential* to improve performance on a task, it is difficult to interpret how effective it was, or diagnose why performance falls short.

We contribute decision-theoretic methods that treat explanations as means to an end: they are valuable insofar as they improve an agent's expected decision performance. Our approach targets three evaluation-relevant estimands:

- **Theoretic Value of Explanation.** Prior to observing human decisions, what is the best case gain in performance that could be attributed to exploiting the available information (as we intend explanations to help human decision-makers do)?

- **Human-Complementary Value of Explanation.** After observing the human baseline decision policy, how much of the theoretically attributable gain is not already captured by human judgment?

- **Behavioral Value of Explanation.** After deploying the explanation, what is the causal effect of providing the explanation to human decision-makers?

Our contributions are: 1) A formalization for specifying decision problems for explanation validation; 2) Decision theoretic upper bounds and estimands for explanation value; 3) Empirical estimators and a corresponding validation workflow; and 4) Demonstrations of the approach to human-AI decision support and mechanistic interpretability.

---

[1]Northwestern University [2]UCSD. Correspondence to: Jessica Hullman <jhullman@northwestern.edu>.

*Proceedings of the 43rd International Conference on Machine Learning*, Seoul, South Korea. PMLR 306, 2026. Copyright 2026 by the author(s).

[1]Code and data are available at https://osf.io/cqbr3/overview?view_only=195c1aabef6445828b6ae8418df3613a

## 2. Decision Problems and the Value of Signals

We assume a decision task, where an explanation of a model's performance is intended to improve the performance of the decision-maker. A model $f : \mathcal{X}_{AI} \to \mathcal{Y}$ predicts a *label* $y \in \mathcal{Y}$ from a feature vector $\boldsymbol{x}_{AI} \in \mathcal{X}_{AI} \subseteq \mathbb{R}^{d_{AI}}$. We assume the predictions of $f$ on test instances where labels are unknown are of interest. Given a test input $\boldsymbol{x}_{AI}$, we denote the model's prediction as $\hat{y} = f(\boldsymbol{x}_{AI}) \in \mathcal{Y}$.

We consider settings where a decision-maker (or *agent*) may be given additional information about the model's behavior in the form of an explanation, which is a function of the features and prediction: $\mathcal{E} : \mathcal{Y} \times \mathcal{X}_{AI} \to \mathcal{Z}$, where $\mathcal{Z}$ is the space of explanations. The structure of $\mathcal{Z}$ depends on the type of explanation—e.g., it could be a set of feature importance or saliency scores, rules, counterfactual or prototype examples, parameters of a transparent surrogate model, or (in mechanistic interpretability), a structured summary of input-conditioned internal computations like latent features or activations, or circuit-level attributions deterministically induced by $\boldsymbol{x}_{AI}$ and $\hat{y}$. Given a model prediction $\hat{y} = f(\boldsymbol{x}_{AI})$, we define the explanation as $z = \mathcal{E}(\hat{y}, \boldsymbol{x}_{AI})$.

In some scenarios where a human decision-maker has access to AI predictions, the decision-maker has access to additional features $\boldsymbol{x}_H \in \mathcal{X}_H \subseteq \mathbb{R}^{d_H}$ beyond the feature representation $\boldsymbol{x}_{AI}$ that the model has access to (e.g., a clinician can directly observe some features of a patient). We denote the full set of features $\boldsymbol{x} = (\boldsymbol{x}_{AI}, \boldsymbol{x}_H)$, from sample space $\mathcal{X} = \mathcal{X}_{AI} \times \mathcal{X}_H$. We represent the human's baseline decision policy absent the explanation as $f^H : \mathcal{X} \times \mathcal{Y} \to \mathcal{A}$, and denote their decisions as $a^H = f^H(\boldsymbol{x}, \hat{y})$.

### 2.1. Decision Tasks

Given a task where we would present an explanation, we specify an associated **decision task** [58], consisting of:

- An *action space* $\mathcal{A}$, a set of actions available to the decision-maker.
- A *state space* $\mathcal{S}$, a set of mutually exclusive states of the world, where the true state of the world is unknown to the decision-maker at decision time.
- A *utility function* (or *scoring rule*) $u : \mathcal{A} \times \mathcal{S} \to \mathbb{R}$ that scores action–state pairs.
- A probability distribution $p$ over $\mathcal{S}$ describing the prior probability of the state.

In practice, the utility function may not be known. In such cases, a class of decision problems corresponding to all problems with functions of a given type can be specified instead of a single problem (see e.g., Appendix D).

We denote the true state at the decision time as $s$, a realization of the random variable $S \in \mathcal{S}$ with distribution $p$. The expected quality of a decision is given by its *expected utility*:

$$u(a, p) = \mathbb{E}_{s \sim p}\left[u(a, s)\right].$$

Any use case can be formalized as a decision task so long as 1) it is possible to define a ground-truth state and 2) there is uncertainty about that state at the time of the decision.

We represent the information available to a decision-maker before they choose an action as an **signal** $V \in \mathcal{V}$. By default, we assume the decision-maker has access to the following explicit information for each decision: the features of the instance $X = \boldsymbol{x}$, the model prediction $\hat{Y} = \hat{y}$, and the explanation $Z = z$.

To evaluate the performance of agents on a decision task requires some labeled data: we evaluate with respect to an *information model* $p \in \mathcal{P}(\mathcal{V} \times \mathcal{S})$. This joint distribution assigns to each possible pairing of signal $V = v$ and state $S = s$ a probability $p(v, s)$. Given the information model, we can derive the prior distribution of the state $p(s)$, where $p(s)$ denotes the probability that $S = s$. We use **decision problem** to refer to the combination of a decision task and an information model.

**Extension to belief formation**  While the benchmarks below are defined directly via $u(a, s)$, this approach also enables studying how explanations help people form accurate beliefs about the state. To do so, one must make use of an equivalent *proper scoring rule*, as only these rules incentivize agents to report their true beliefs (as they cannot obtain a higher score by deviating) [25]. For any utility function $u : \mathcal{A} \times \mathcal{S} \to \mathbb{R}$, there is an equivalent proper scoring rule where the action space is a probabilistic belief, i.e., $\hat{u}(p, s) = u(\arg\max_{a \in \mathcal{A}} \mathbb{E}_{s' \sim p}\left[u(a, s')\right], s)$.

---

**Example** (Medical Decision Making). *A physician decides whether a patient should undergo a biopsy.*

- *State $s \in \{0, 1\}$, whether the disease is present.*
- *Action $a \in \{0, 1\}$, whether to conduct an invasive biopsy procedure.*
- *Utility Function*

$$u(a, s) = \begin{cases} 1, & \text{if } a = 1, s = 1 \\ 0, & \text{if } a = 1, s = 0 \\ \epsilon, & \text{if } a = 0 \end{cases}$$

*where $s$ is unobservable when $a = 0$ and $\epsilon \in (0, 1)$ is a constant utility when no biopsy is conducted thus the state cannot be observed.*

- *Signals $V = X \cup \hat{Y} \cup Z$, information about the patient (e.g., a chest radiograph), model prediction (e.g., a risk score predicted by a computer vision model), and an explanation (e.g., saliency-based methods such as LIME and SHAP, or example-based methods such as*

*nearest-neighbor factual or counterfactual examples).*

*If the researcher wants to study how explanations affect beliefs about the disease, they can translate the problem into a proper scoring rule:* $\hat{u}(p,0) = \epsilon \times \mathbf{1}_{p<\epsilon}$ *and* $\hat{u}(p,1) = \mathbf{1}_{p\geq\epsilon} + \epsilon \times \mathbf{1}_{p<\epsilon}$.

*Appendix A provides further examples of decision tasks.*

## 2.2. Value of Information

Given a decision problem, we can quantify the maximum value of any piece of information to performance on that problem using the conceptual device of a Bayesian rational agent. The theoretic value of a signal is the improvement in this best case agent's expected utility relative to lacking the signal. Assuming a utility maximizing Bayesian learner with oracle access to the information model $p \in \mathcal{P}(\mathcal{V} \times \mathcal{S})$, upon observing a signal $v$, this agent uses their knowledge of $p$ to Bayesian update from the prior $p(s) = \sum_{v \in \mathcal{V}} p(v,s)$ to *posterior beliefs* for each state $s \in \mathcal{S}$:

$$p(s \mid v) = \frac{p(v,s)}{\sum_{s \in \mathcal{S}} p(v,s)} \quad (1)$$

They select the *utility-maximizing action* under the beliefs. The expected performance over the information model is:

$$\mathrm{R}_V := \mathbb{E}_{v \sim p(v)} \left[ \max_{a \in \mathcal{A}} \mathbb{E}_{s \sim p(s|v)}[u(a,s) \mid V = v] \right] \quad (2)$$

where we refer to $\mathrm{R}(\cdot)$ as the **rational agent benchmark**: the expected performance over the information model of taking the utility-maximizing action after observing the signal. As a convenience for later defining the value of an explanation over some other signal, we will use $\mathrm{R}_{V_1 \cup V_2}$ to represent the expected performance of the rational agent who observes a combination of $V_1$ and $V_2$, i.e.,

$$\mathrm{R}_{V_1 \cup V_2} := \mathbb{E}_{v \sim p(\cdot)} \left[ \max_{a \in \mathcal{A}} \mathbb{E}_{s \sim p(\cdot)}[u(a,s) \mid V_1 = v_1, V_2 = v_2] \right]$$

The rational agent benchmark *upper bounds* the expected performance of any agent restricted to the same signals. Thus, the benchmark provides a target against which we can compare the performance of behavioral agents that we observe in practice.

To quantify the theoretic value of a signal, we compare the rational agent benchmark to the best performance of any agent who ignores the signals entirely. We refer to the latter performance as the *rational agent baseline* following [66]:

$$\mathrm{R}_{\varnothing} = \max_{a \in \mathcal{A}} \mathbb{E}_{s \sim p(\cdot)}[u(a,s)] \quad (3)$$

**Definition 1.** *The difference between the rational agent benchmark and rational agent baseline is the **value of information** $\Delta$ of the signal:* $\Delta = \mathrm{R} - \mathrm{R}_{\varnothing}$

## 3. Theoretic Value of Explanation

We first characterize if, prior to observing human use of explanations, an explanation has *potential* to improve performance on a decision problem. The estimand we target for validating new explanation or interpretability techniques is the expected improvement in performance of decision-makers who go from having access to a reduced signal of the form $v_{\neg \mathcal{E}} = \{\boldsymbol{x}, \hat{y}\}$ to a full signal containing the explanation $v = \{\boldsymbol{x}, \hat{y}, z\}$. As we show below, prior to deploying an explanation method, we can use Definition 1 to quantify the best-case potential of the explanation to improve performance on the decision problem(s) of interest.

**Upper Bound**  Given that explanations are expected to help human decision-makers, we might expect that removing the explanation component from the benchmark corresponding to the performance of the rational Bayesian agent with the full signal $\mathrm{R}_{X \cup \hat{Y} \cup Z}$ would *reduce* the rational agent's score, i.e., $\mathrm{R}_{X \cup \hat{Y}} \leq \mathrm{R}_{X \cup \hat{Y} \cup Z}$. However, in Proposition 1, we show that this is not the case, thus offering any explanation $Z = \mathcal{E}(X, \hat{Y})$ to human decision-makers assumes irrationality.

**Proposition 1.** *Given a set of features $X$, a model prediction $\hat{Y}$, and an explanation $Z$ generated by a function taking as input features and model prediction (Section 2), gaining access to the explanation does not improve the expected performance of the idealized agent, i.e.,*

$$\mathrm{R}_{X \cup \hat{Y} \cup Z} = \mathrm{R}_{X \cup \hat{Y}} \quad (4)$$

We prove Proposition 1 by leveraging Blackwell's informativeness theorem [7]. See Appendix B for the proof.

**Corollary 1.** *When a set of features $X$ contains all the input of the model $f$, i.e., $X_{AI} \subseteq X$, gaining access to the model prediction does not improve the expected performance of the idealized agent when they already have access to $X$, i.e.,*

$$\mathrm{R}_{X \cup \hat{Y}} = \mathrm{R}_X$$

Proposition 1 arises due to the fact that the explanations generated by the function $\mathcal{E}$ represent a "garbling" [45] of the model prediction $\hat{Y}$ and features $X$; i.e., at best they are equally informative. Similarly, given access to $X$, the rational agent gains no additional value from the prediction $\hat{Y}$. This is in sharp contrast with our expectations about human decision-makers, who we *do* expect to behave differently with access to an explanation. By believing that explanations are helpful to humans, we are assuming that their expected performance deviates from idealized use of the instance-level information, i.e., we do not necessarily expect people to be able to extract all of the information that is carried by the features without the explanation. In

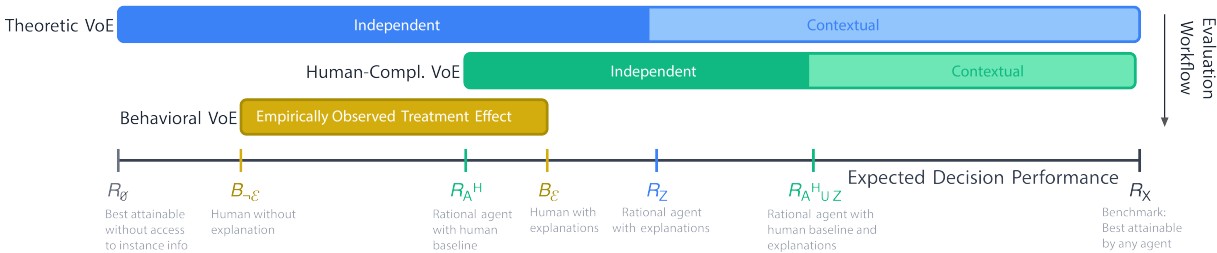

**Figure 1.** Quantities defined in our framework. The researcher can confirm that an explanation has potential by proceeding from the theoretic value of explanation (VoE) to the human-complementary value of explanation to the behavioral value of explanation, comparing the estimates at lower levels to those above.

Appendix E, we offer formal characterizations for how explanations that do not directly convey information on the state can help various kinds of boundedly-rational agents.

That explanations are redundant with the features for a rational agent but can help "unlock" contextual information for irrational decision-makers means that the best-case expected performance with the features $\boldsymbol{x}$ upper bounds the performance of any agent, human or otherwise, with the signal $v = \{\boldsymbol{x}, \hat{y}, z\}$:

**Definition 2.** *The **benchmark** is the expected performance of the rational agent with the features $X$:*

$$\mathrm{R}_X := \mathbb{E}_{\boldsymbol{x} \sim p(\cdot)} \left[ \max_{a \in \mathcal{A}} \mathbb{E}_{s \sim p(\cdot)} [u(a, s) \mid X = \boldsymbol{x}] \right]$$

**Definition 3.** *The **theoretic value of explanation** $\Delta_{\mathcal{E}}$ is the difference in the benchmark and the baseline expected score of the rational Bayesian agent when they have access to only the prior: $\Delta_{\mathcal{E}} = \mathrm{R}_X - \mathrm{R}_{\varnothing}$.*

The theoretic value of explanation describes the boost in performance on the decision problem that we expect in the best case where the explanation helps the human better extract or apply all of the information about the state conveyed by the features. Note that $\Delta_{\mathcal{E}}$ is *not* explanation specific: it depends only on the target decision problem. This is intentional, and demonstrates that based on how we expect explanations to work, the first step in successful explanation validation does not hinge on the explanation at all: it necessitates studying problems where explanations have the possibility of being important.

> **Example** (Medical Treatment). *We use the information model estimated from the MIMIC-IV and MIMIC-CXR datasets [33], with $\epsilon = 0.5$ for demonstration purposes[a]. Because the features $X$ include high-dimensional signals (such as radiographs), we use a coarsening algorithm (Algorithm 1) to prevent overfitting.*
>
> *The theoretic value of explanation $\Delta_{\mathcal{E}}$ is $\mathrm{R}_X - \mathrm{R}_{\varnothing} \simeq$ 0.12, or roughly 25% of the baseline, suggesting potential for explanations to help.*

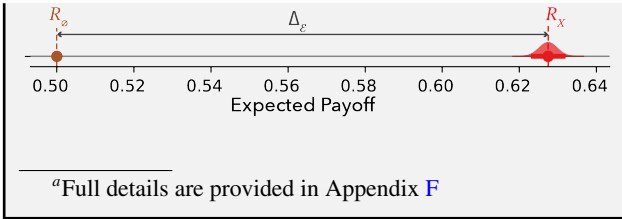

_______________
[a]Full details are provided in Appendix F

**Decomposition of $\Delta_{\mathcal{E}}$ into component values** $\Delta_{\mathcal{E}}$ creates a span along which the value of component signals–including the explanation $Z$ and independent human decisions $A^H$–can be contrasted. To compare specific explanation methods, we can decompose $\Delta_{\mathcal{E}}$ as the sum of two quantities. The **independent** theoretic value of explanation $\Delta_{\text{ind-}\mathcal{E}}$ is the improvement in performance that can be achieved from the information about the state conveyed directly by the explanation. The **contextual** theoretic value of explanation $\Delta_{\text{cont-}\mathcal{E}}$ is the additional improvement that can be achieved by obtaining information about the state conveyed by the features $X$.

**Definition 4.** *$\Delta_{\text{ind-}\mathcal{E}}$ is the expected change in score of the rational Bayesian agent when they have access to the explanation $Z$ versus only the prior $\Delta_{\text{ind-}\mathcal{E}} = \mathrm{R}_Z - \mathrm{R}_{\varnothing}$.*

**Definition 5.** *$\Delta_{\text{cont-}\mathcal{E}}$ is the expected change in the score of the rational Bayesian agent when they have access to the features $X$ versus the explanation $Z$ : $\Delta_{\text{cont-}\mathcal{E}} = \mathrm{R}_X - \mathrm{R}_Z$.*

Decomposing $\Delta_{\mathcal{E}}$ makes it possible to compare, in the spirit of prior attempts to estimate the value of explanations [14], the a priori value of the explanation component of the signal alone as a proportion of the benchmark. In doing so, however, it should be noted that the ranking of these independent values does not capture the full potential of the explanations, and therefore may not predict their effectiveness in practice.

> **Example** (Medical Treatment). *We train a predictive model based on a radiology foundation model [59], using the MIMIC-IV [33]. We generate four types of explanations: LIME [57], SHAP [43], factual example (the nearest-neighbor instance with the same predictive la-*

*bel), and counterfactual example (the nearest-neighbor instance with a different predictive label). We coarsen the explanations using Algorithm 1.*

*The independent theoretic value of explanation $\Delta_{ind\text{-}\mathcal{E}}$ varies with different explanation techniques. The two example-based explanations have larger independent theoretic value ($R_Z - R_\varnothing$) than the salience-based ones (LIME and SHAP). However, the independent theoretic values of the explanations are substantially lower than the overall theoretic value $\Delta_\mathcal{E}$. Hence even with the most independently informative explanation, the agent must extract decision-relevant information from the features to achieve the benchmark (i.e., all $\Delta_{cont\text{-}\mathcal{E}}$ are large).*

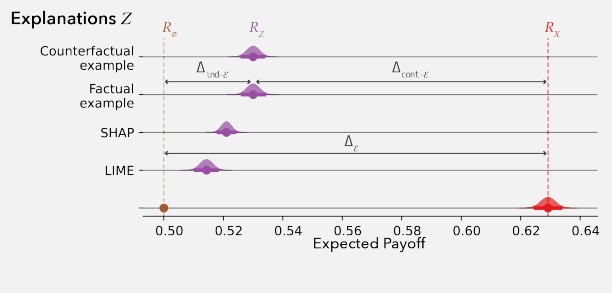

### 3.1. Estimating $\Delta_\mathcal{E}$ in Practice

**Coarsened information model**  $R_X$ and $\Delta_\mathcal{E}$ depend on the joint distribution $p$, which we estimate from an evaluation dataset $D = \{(s_i, \hat{y}_i, z_i, \boldsymbol{x}_i)\}_{i=1}^T$. In low-dimensional settings, $p$ can be estimated directly. However, when $X$ or $Z$ are high-dimensional (e.g., images, text,), the plug-in rational benchmark in Equation (2) can overfit, yielding spuriously perfect benchmark performance. Overfitting can be avoided by identifying a coarsened signal structure that aggregates raw signals into equivalence classes. Concretely, we learn clustering maps $\mathcal{C}_X : \mathcal{X} \rightarrow [K_{\boldsymbol{x}}]$ and $\mathcal{C}_Z : \mathcal{Z} \rightarrow [K_Z]$ and compute the empirical posterior $\hat{p}(V|\mathcal{C}_X(X))$ on a training split $\mathcal{D}_{tr}$, then evaluate on a held-out split $\mathcal{D}_{test}$. We select the coarsening that maximizes held-out rational performance subject to a small train–test gap constraint (see Appendix C). Given that $\hat{Y}$ and $Z$ cannot be more informative than the instance-level signal by construction, coarsening should not invert this ordering, which we ensure by restricting the search over $\mathcal{C}_X$ and $\mathcal{C}_Z$ to those that preserve the garbling relationships. Under coarsening, the behavioral value of explanation (Def. 9) should be estimated with the coarsened representation.

**Ambiguity in decision problem specification**  Whenever there is ambiguity about how to best define the decision problem, a robust analysis approach that defines a class of decision problems (e.g., [27]) can be used in calculating the theoretic value of explanation. Because every payoff function can be translated into a proper scoring rule, this entails

doing a worst-case analysis over a grid of proper scoring rules. The $\Delta_\mathcal{E}$ and $\Delta_{\mathcal{E}_{compl}}$ estimated under this approach can be used to approximate the Blackwell order of the explanations, i.e., for any decision problem, the explanation with higher $\Delta_\mathcal{E}$ gives higher decision-relevant information. We provide full definitions and theorems in Appendix D.

**Accounting for unobserved human private information**
A second challenge is that treating $R_X$ as an upper bound assumes knowledge of $X$, including all decision-relevant information available to the human. In practice, we may not know if they have private features beyond those known to the model. In such cases, we can treat the baseline human decisions $A^H$ as a behavioral proxy that approximately summarizes the information the human uses, in the spirit of information-economics "revelation through action" assumptions common in signaling models (e.g., [61]). In practice, we test whether $X_{AI}$ is sufficient by comparing $R_{X_{AI}}$ to $R_{X_{AI} \cup A^H}$. If $R_{X_{AI} \cup A^H} > R_{X_{AI}}$, then $R_{X_{AI} \cup A^H}$ is the appropriate upper bound.

## 4. Human-Complementary and Behavioral Value of Explanation

We define two additional estimands (with associated estimators) that take into account human baseline decisions and performance with the explanation.

### 4.1. Human-Complementary Value of Explanation

An explanation can only provide or "unlock" information about the state for a human if they have not already exploited that information in the decisions. Although we focus on humans because explanations have conventionally been designed for human understanding and human decision support, the same estimand can be defined with respect to any baseline agent whose decisions the explanation is intended to improve upon. After establishing the theoretic potential of explanations for a problem, we can check for human-complementary potential by eliciting baseline human decisions $A^H$. We use these to upper bound the value of the decision-relevant information contained in the decisions by offering them to a rational agent in place of $X$, creating a calibrated baseline human benchmark: $R_{A^H}$. We then condition on this calibrated information to estimate the additional marginal value of the explanation over information already available to the decision-makers.

**Definition 6.** *The **potential complementary value of explanation** $\Delta_{\mathcal{E}_{compl}}$ is the expected improvement in the performance of the rational Bayesian agent when they have access to the features $X$ for each instance and human baseline decisions $A^H$ versus when they lack access to $X$:*
$$\Delta_{\mathcal{E}_{compl}} = R_X - R_{A^H}.$$

Like the theoretic value of explanation $\Delta_{\mathcal{E}}$, we can decompose the potential human-complementary value of explanation into two quantities:

**Definition 7.** *The **independent** potential complementary value of explanation $\Delta_{ind\text{-}\mathcal{E}_{compl}}$ is the expected change in score of the rational Bayesian agent when they have access to the human decision $A^H$ and the explanation $Z$ versus when they lack access to $Z$: $\Delta_{ind\text{-}\mathcal{E}_{compl}} = \mathrm{R}_{A^H \cup Z} - \mathrm{R}_{A^H}$.*

**Definition 8.** *The **contextual** potential complementary value of explanation $\Delta_{cont\text{-}\mathcal{E}_{compl}}$ is the expected change in the score of the rational Bayesian agent when they have access to the features $X$ versus the human decision $A^H$ and the explanation $Z$: $\Delta_{cont\text{-}\mathcal{E}_{compl}} = \mathrm{R}_X - \mathrm{R}_{A^H \cup Z}$.*

While $\Delta_{\mathcal{E}_{compl}}$ gives a sense of how much complementary information is contained in a decision problem, $\Delta_{ind\text{-}\mathcal{E}_{compl}}$ and $\Delta_{cont\text{-}\mathcal{E}_{compl}}$ describe how human-complementary information is distributed between the explanation and the features.

---

**Example** (Medical Treatment). *We induce human decisions for the medical example (i.e., whether to conduct a biopsy) by applying a rule-based model to the radiology reports in MIMIC-CXR [33].*

*While the features have large human-complementary information value ($\Delta_{\mathcal{E}_{compl}} \simeq 0.12$), the explanations do not offer much human-complementary information on their own (i.e., all $\Delta_{ind\text{-}\mathcal{E}_{compl}}$ are low). The human decisions offer complementary information over the explanations (i.e., all $\mathrm{R}_{A^H \cup Z}$ are higher than the corresponding $\mathrm{R}_Z$).*

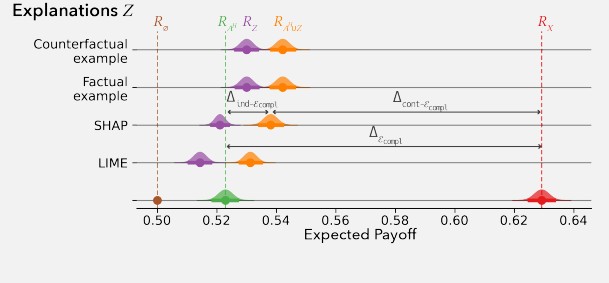

---

### 4.2. Behavioral Value of Explanation

After establishing that an explanation method offers theoretic and potential human-complementary value for the decision problem(s) at hand, the next step is to evaluate its effect on human decision-makers. Under randomized assignment of explanations, the canonical approach quantifies this via the average treatment effect (ATE), the average expected difference in outcomes with and without the intervention [5]. Let $p^B$ be the joint distribution over the human decisions and state, i.e., $p^B \in \mathcal{P}(\mathcal{A} \times \mathcal{S})$ when human agents have access to "full" signals that include the explanation and AI prediction ($v = \{\boldsymbol{x}, \hat{y}, z\}$). Let $\mathrm{B} = \mathbb{E}_{(a,s) \sim p^B} u(a, s)$ be the expected utility. Let $p^B_{\neg \mathcal{E}}$ be the joint distribution over

the human decisions and state when human agents have the same signal minus the explanation ($v = \{\boldsymbol{x}, \hat{y}\}$), with associated expected utility $\mathrm{B}_{\neg \mathcal{E}} = \mathbb{E}_{(a,s) \sim p^B_{\neg \mathcal{E}}} u(a, s)$.

**Definition 9.** *The **behavioral value of explanation** $\Delta_{\mathcal{E}_{behavioral}}$ is the difference in expected score of a human decision-maker when they have access to the explanation versus when they do not: $\Delta_{\mathcal{E}_{behavioral}} = \mathrm{B} - \mathrm{B}_{\neg \mathcal{E}}$*

It may be possible to estimate the behavioral value of explanation $\Delta_{\mathcal{E}_{\text{behavioral}}}$ from raw study results in some cases, but typical behavioral study designs necessitate isolating the effect of the explanation by fitting a structural statistical model (e.g., a multiple regression) that controls for confounding factors like trial order or individual differences [67]. See Guo et al. [26] for examples of fitting such models to studies on human reliance on AI models.

## 5. Applying the Framework in Practice

The three values should be interpreted as a sequence of tests.

**Step 1: Does the problem provide sufficient theoretic value of explanation?** First, define the decision problem of interest and calculate the theoretic value, $\Delta_{\mathcal{E}} = \mathrm{R}_X - \mathrm{R}_{\varnothing}$. This quantity describes how much potential there is for an explanation to improve task performance. It is an upper bound on the value available from helping an agent extract all relevant information from the instance representation. The main diagnostic question it answers is **whether a decision problem is worth studying**. If $\Delta_{\mathcal{E}}$ is negligible, then the decision problem offers little room for any explanation to improve performance: even an idealized agent that fully exploits the instance representation would not do much better than the best fixed decision based on the prior. In this case, explanation evaluation is unlikely to be informative, because there is little decision-relevant information for an explanation to unlock.

**Step 2: Check the human-complementary value of explanation.** Next, collect baseline (independent) human decisions[2] and calculate the human-complementary value, $\Delta_{\mathcal{E}_{\text{compl}}} = \mathrm{R}_X - \mathrm{R}_{A^H}$. This quantity describes how much potential there is for an explanation to improve performance beyond what is already implied by baseline human judgments. The main diagnostic question this answers is **whether there is room for explanations to help decision-makers do better than we would expect them to do without the additional support**. If humans without the explanation are already close to the best possible performance on the task, then there is little reason to expect explanations to produce a large improvement. This quantity therefore targets the complementarity of the information that explanations can help communicate to what human decision-makers already

---

[2]This step can be used to compare candidate explanation methods before running a full behavioral study.

use.

**Step 3: Estimate and interpret the behavioral value of explanation.** Only after the theoretic and human-complementary checks indicate meaningful potential should one proceed to deploy the AI tools and conduct an evaluation to estimate the behavioral value, $\Delta_{\mathcal{E}_{\text{behavioral}}} = \text{B} - \text{B}_{\neg\mathcal{E}}$. The magnitude of the behavioral value can be interpreted relative to the theoretic and human-complementary values. Comparing $\Delta_{\mathcal{E}_{\text{behavioral}}}$ to $\Delta_{\mathcal{E}}$ shows how much of the total available value is realized by the behavioral agents. Comparing $\Delta_{\mathcal{E}_{\text{behavioral}}}$ to $\Delta_{\mathcal{E}_{\text{compl}}}$ shows how much of the value not already captured by baseline human judgment is realized after providing the explanation. Comparing $\text{B}$ to $\text{R}_X$ shows how close participants with explanations come to the rational benchmark. This comparison also helps diagnose failure modes. If $\Delta_{\mathcal{E}}$ is large but $\Delta_{\mathcal{E}_{\text{compl}}}$ is small, then explanations may have little room to improve over baseline human decisions. If both $\Delta_{\mathcal{E}}$ and $\Delta_{\mathcal{E}_{\text{compl}}}$ are large but $\Delta_{\mathcal{E}_{\text{behavioral}}}$ is small, then the explanation contains or could unlock useful information, but participants are not using it effectively.

Note that $\Delta_{\mathcal{E}_{\text{behavioral}}}$ may *exceed* $\Delta_{\mathcal{E}}$, when behavioral participants do worse than the rational agent with only the prior $(\text{R}_{\varnothing})$ without explanations, but extract significant information given access. Such results indicate poor task design, as simply conveying the prior could improve performance [30]. In that case, the first intervention to try is to communicate the prior or base rate, rather than to introduce an explanation.

Note also that when human decisions with explanations are unavailable, theoretic and human-complementary values can still be useful proxies. However, they should be interpreted as best-case quantities. A large value of $\Delta_{\mathcal{E}}$ or $\Delta_{\mathcal{E}_{\text{compl}}}$ relative to $\text{R}_{\varnothing}$ indicates potential for improvement, but it does not imply that decision-makers will realize that value in practice. Behavioral evaluation remains necessary to estimate how much of the available value explanations actually deliver.

# 6. Demonstrations

We apply our framework to two common use cases: explanations as decision support in human–AI studies, and mechanistic interpretability for identifying threats to model behavior. For each, we specify the decision task (state, signals, actions, utilities; Section 2) and estimate benchmarks.

## 6.1. Retrospective Analysis of Human–AI Studies

We reanalyze two controlled studies where an AI assists a human decision-maker: deceptive text detection [38] and sentiment classification of text [6].

|  | Heatmap | Examples | Random Heatmap |
|---|---|---|---|
| Deception Detection | 0.10 [0.07, 0.13] | 0.08 [0.04, 0.11] | 0.07 [0.03, 0.10] |

|  | Explain-top-1 | Explain-top-2 | Adaptive | Expert |
|---|---|---|---|---|
| Amzbook | 0.01 [0.00, 0.02] | 0.00 [-0.01, 0.014] | 0.01 [0.00, 0.02] | 0.00 [-0.01, 0.01] |
| Beer | -0.01 [-0.02, 0.00] | -0.02 [-0.03, -0.01] | -0.01 [-0.02, 0.00] | 0.00 [-0.01, 0.01] |

**Table 1.** Behavioral value of explanation and 95% Confidence Intervals for three explanation types in the deception detection task and four explanation types in the sentiment classification tasks (Amazon Books and Beer datasets). The behavioral value of explanation is the average difference in accuracy between the human decision-maker with and without the explanation.

**Decision tasks and signals.** In both tasks, $s \in \{0, 1\}$ is the true label, and action $a \in \{0, 1\}$ is a binary classification, both corresponding to whether a hotel review is deceptive or a product review expresses positive sentiment. $X$ is the instance text. In deception detection, following Lai & Tan [38], we study three explanation conditions: two nearest-neighbor instances (*Example*), feature-attribution heatmaps generated with LIME (*Heatmap*), and random heatmaps that randomly highlight words (*Random Heatmap*). In sentiment classification, following Bansal et al. [6], we study four explanation types: expert-generated explanations (*Expert*), explanation for the label with higher confidence (*Explain-top-1*), explanation for both labels (*Explain-top-2*), and an adaptive explanation condition that dynamically selects between the two (*Adaptive*). We also analyze the original study's control condition showing the model predicted confidence. Full details are in Appendix F.

**Theoretical and human-complementary value of explanation** Figure 2 summarizes theoretical values (with sentiment split by dataset). For deception detection, Example and Heatmap explanations offer substantial independent theoretical value ($\Delta_{\text{ind-}\mathcal{E}} \simeq 0.4 - 0.5\Delta_{\mathcal{E}}$. They offer moderate independent human-complementary value (e.g., $\Delta_{\text{ind-}\mathcal{E}_{\text{compl}}} \simeq 0.26 - 0.39\Delta_{\mathcal{E}_{\text{compl}}}$. However, most potential value of effective explanation use comes from their potential to unlock information in the features, which can improve participants' accuracy by roughly 10% and 15%, respectively, over performance with only the AI prediction.

For sentiment classification, explanations vary in independent theoretical value (e.g., $\Delta_{\text{ind-}\mathcal{E}} \simeq 0.5\Delta_{\mathcal{E}}$ for the *Explain-top-1* explanations in Amzbook), but offer little independent human-complementary value ($\Delta_{\text{ind-}\mathcal{E}_{\text{compl}}} \simeq 0$ for all types but *Explain-top-1* in Books).If applied before running the study, the framework would have predicted explanations would not be effective for this task, as results below show.

**Interpreting behavioral effects** Table 1 shows the behavioral value for explanation types in both studies (see also Figure 2). For deception detection, all three explanations reliably improve participants' decision performance. This

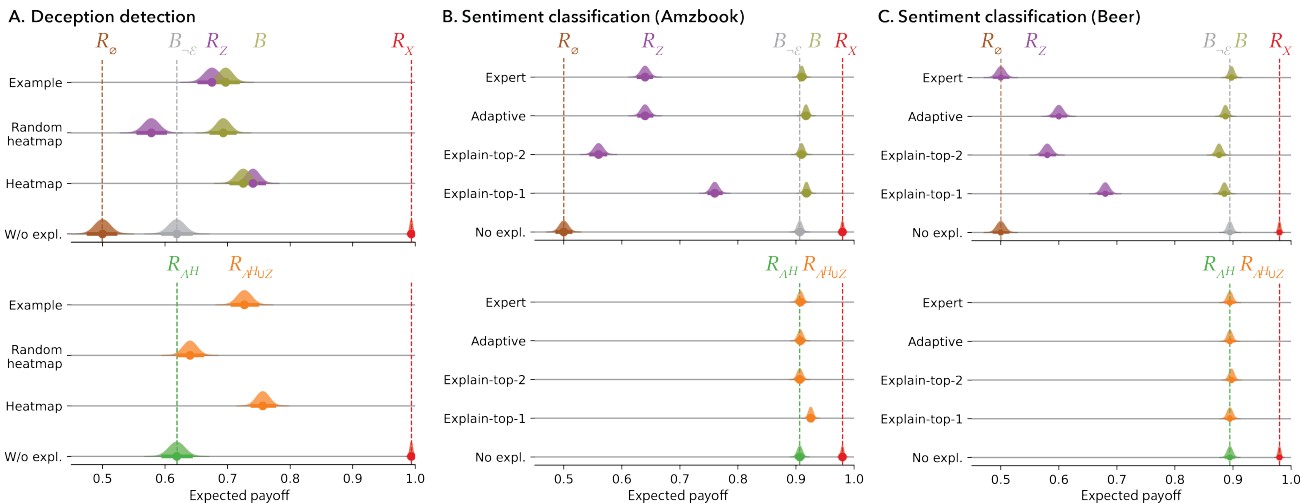

**Figure 2.** Re-analysis of human-AI decision support in two prior studies [6, 38]. Top: theoretic benchmarks (Section 3) against behavioral values (Section 4.2). Bottom: potential complementary value (Section 4). Error bars give bootstrapped 95% CIs (N=1000).

is congruent with the baseline human decisions without the explanation offering limited information over the prior (i.e., $R_{A^H} - R_{\varnothing} \simeq 0.1 \simeq 0.2\Delta_{\mathcal{E}}$): humans were not very good at judging deception without the explanation.

In contrast, for sentiment classification, our analysis suggests it is not very surprising that participants did not improve much when offered explanations (i.e., $R_{A^H \cup Z}$ is very close to $R_{A^H}$ for both datasets), because none of the explanations offered much human-complementary information about the state.

Overall, we find the tasks differ substantially in how much value explanations offer over what participants can do without them, helping explain the study results. By piloting on a sample of participants *without* explanations first, explanations could instead be selected to maximize human-complementary information, increasing knowledge gain.

### 6.2. Mechanistic Interpretability Alignment Audit

We apply the framework to an alignment-audit task inspired by Marks et al. [44], where a mechanistic interpretability technique is used to detect a problematic training bias.

**Decision tasks and signals.** The state $s \in \{0, 1\}$ corresponds to whether the model was trained on toxic comments. We train two transformers on Jigsaw toxic comment classification dataset splits (with vs. without toxic comments) [17]. The action $a \in \{0, 1\}$ is a binary classification of the state, and utility is defined as accuracy, i.e., $u(a, s) = \mathbf{1}_{a=s}$. Signals include the input instance and the explanation generated on the SAE features. We use sparse autoencoder (SAE)-based explanations following Marks et al. [44]: *SAE-top-1* to *SAE-top-5* for explanations displaying the top 1 to 5 most important SAE features. We use Qwen3-14B to

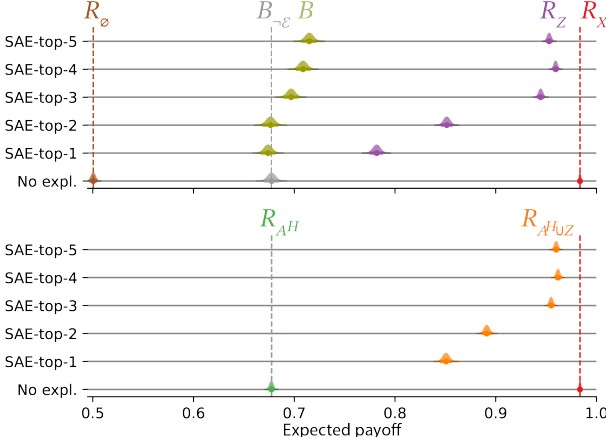

**Figure 3.** Alignment audit results. Theoretic value increases up to 3 SAE features then plateaus. Behavioral value increases with the number of features, but indicates substantial room to improve relative to the benchmarks.

simulate human decisions by prompting it to judge whether the model is trained with toxicity bias or not. Full details are in Appendix F.

**Theoretical and human-complementary value of explanation.** The theoretic value of SAE explanations increases from *SAE-top-1* to *SAE-top-3* then plateaus; by *SAE-top-3*, the independent theoretic value is close to the full explanation value (i.e., the remaining gap to benchmark $\Delta_{\text{cont-}\mathcal{E}} \simeq 0.04\Delta_{\mathcal{E}}$ for *SAE-top-3* to *SAE-top-5*). Hence SAE explanations alone offer enough decision-relevant information to help decision-makers judge whether the model is trained on toxic comments, without requiring decision-makers to exploit information in the features. Human-

| SAE-top-1 | SAE-top-2 | SAE-top-3 | SAE-top-4 | SAE-top-5 |
|---|---|---|---|---|
| 0.00 [-0.01, 0.01] | 0.00 [-0.01, 0.07] | 0.02 [0.01, 0.03] | 0.03 [0.02, 0.04] | 0.04 [0.03, 0.05] |

**Table 2.** Average Treatment Effect (ATE) and 95% Confidence Interval for five explanation types in the alignment audit task.

complementary value is also substantial across SAE explanations (i.e., more than half of the human-complementary value of the explanation is offered by the SAE explanations, $\Delta_{\text{ind-}\mathcal{E}_{\text{compl}}} > 0.5\Delta_{\mathcal{E}_{\text{compl}}}$).

**Interpreting behavioral effects.** Table 2 shows behavioral value increases with the number of SAE features. Since theoretic value plateaus after *SAE-top-3*, the remaining gap to the benchmark suggests under-extraction of available information, motivating further interventions (e.g., training) to improve use.

## 7. Related Work

Explanations are designed to improve trust, model reliance, and understanding, yet empirical studies find little evidence that explanations improve decisions or reliance [6, 11, 12, 26, 38], as corroborated by meta-analysis [64]. Prior work attributes this to design issues, including non-uniqueness and potential contradictions among explanations for the same prediction [9, 19, 34, 37, 46]. Our approach shifts focus from formal analyses of independent properties of explanations like faithfulness [see e.g., 4, 23, 55, 63]–which are neither necessary nor sufficient for explanations to improve human understanding [54]–to the theoretic potential explanations have to improve performance on concrete decision tasks, squarely addressing ambiguity about attainable performance in a scenario noted by prior authors [26, 36, 42, 56]. The estimands and corresponding workflow we contribute can be seen as a direct response to recent calls for more "actionable" or pragmatic approaches to explanation and interpretability techniques [10, 21, 52, 60] connected to performance on concrete tasks.

Definition 3 can be contrasted with prior notions of explanation value. Chen et al. [14] use the predictive accuracy that can be attained from the information explanations convey for a use case. Our framework clarifies that an explanation's value in isolation does not reflect its full potential: less directly decision-informative signals (e.g., instance-invariant accuracy summaries) can be more useful if they steer people to better use private information (e.g., by indicating low model reliability). Our work provides a complementary formal lens on previous discussion of the role of "intuition" in explanation use [13, 15]. It helps resolve debates about crucial properties for explanations to improve decision performance like verifiability. While Fok & Weld [22] argue that for explanations to aid decisions, they must be "verifiable"–

meaning they allow a decision maker to verify the AI recommendation, we show otherwise: verifiability corresponds to usefulness only insofar as the explanation directly reveals state—an assumption that ignores correlations among explanations, predictions, and features by construction. Our decomposition of $\Delta_{\mathcal{E}}$ enables testing when "verifiability" correlates with effectiveness (Section 4.2).

## Impact Statement

Our work advances the field of explainability and machine learning, which stands to contribute to a number of public-facing and scientific domains. To the best of our knowledge, there are no particular negative social consequences imposed by our work compared to machine learning research in general.

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

## A. Additional Example Decision Problems

**Example** (Model Debugging). *Given a training dataset $\{(\boldsymbol{x}_i, y_i)\}_{i \in [N]}$ and a model $f : \mathcal{X} \to \mathcal{Y}$, a developer uses explanations to determine how to improve a model's expected performance on a test dataset $\{\boldsymbol{x}_i^{new}\}_{i \in [N']}$.*

- *State: $s = (y_i^{new}) \in \mathcal{Y}^{N'}$, the labels of the text dataset.*
- *Action: $a \in \{f : \mathcal{X} \to \mathcal{Y}\}$, the debugged model.*
- *Utility: $u(a, s) = \mathbb{E}_{(\boldsymbol{x}_i^{new}, y_i^{new})} \left[ \mathbf{1}_{a(\boldsymbol{x}_i^{new}) = y_i^{new}} \right]$, the test performance of the debugged model.*
- *Signal: $V = X \cup \hat{Y} \cup Z$. $X$ represents the test dataset, $\{\boldsymbol{x}_i^{new}\}_{i \in [N']}$. $\hat{Y}$ represents the model predictions on the test dataset, $\{f(\boldsymbol{x}_i^{new})\}_{i \in [N']}$. $Z$ represent the explanations of the model prediction such as LIME, SHAP, or sparse autoencoder.*

**Example** (Model Auditing [44]). *A researcher uses sparse autoencoders (SAEs) to identify what caused an LLM to exhibit an unwanted behavior, such as a systematic error $e$ affecting training data that leads to user-sycophancy.*

- *State: $s \in \{f : \mathcal{X} \to \mathcal{Y}\}$, the space of models trained on a dataset with bias $e^* \in E$.*
- *Action: $a \in \mathcal{A} = E$, the bias the researcher identifies in the training data.*
- *Utility: $u(a, s) = \mathbf{1}_{a = e^*}$, whether the bias is correctly identified.*
- *Signal: $V = X \cup \hat{Y} \cup Z_{SAE}$, including the prompt/input, the model prediction, and the sparse autoencoder (SAE) interpretation.*

## B. Proof of Proposition 1

*Proof.* Our proof is based on the Blackwell's informativeness theorem.

**Theorem 2** ((Informal proof) Blackwell's informativeness theorem [7]). *Given a decision task, let $V_1$ and $V_2$ be two random variables, with the conditional probabilities as $\sigma_1 = p(v_1 \mid s)$ and $\sigma_2 = p(v_2 \mid s)$ respectively. If there exists a function $f$ such that $\sigma_2 = f(\sigma_1)$, then $\mathrm{R}_{V_1} \geq \mathrm{R}_{V_2}$.*

We prove Proposition 1 by showing that there exists a function $f_1$ such that $p(\boldsymbol{x}, \hat{y}, z \mid s) = f_1(p(\boldsymbol{x}, \hat{y} \mid s))$ and a function $f_2$ such that $p(\boldsymbol{x}, \hat{y} \mid s) = f_2(p(\boldsymbol{x}, \hat{y}, z \mid s))$.

The first function $f_1$ can be constructed using the fact that explanations are deterministic functions of the features and model prediction, i.e., $Z = \mathcal{E}(\hat{Y}, X_{AI})$. Therefore, the conditional distribution of the explanation given $X$ and $\hat{Y}$ is deterministic, i.e., $p(z \mid \boldsymbol{x}, \hat{y}, s) = \mathbf{1}_{z = \mathcal{E}(\boldsymbol{x}, \hat{y})}$. Then, we can construct the function $f_1$ by Bayes' rule: $p(\boldsymbol{x}, \hat{y}, z \mid s) = p(z \mid \boldsymbol{x}, \hat{y}, s) \cdot p(\boldsymbol{x}, \hat{y} \mid s)$.

The function $f_2$ is straightforward to construct by calculating the conditional marginal distribution of $X$ and $\hat{Y}$ given $s$ over the joint distribution $p(\boldsymbol{x}, \hat{y}, z \mid s)$, i.e., $p(\boldsymbol{x}, \hat{y} \mid s) = \sum_z p(\boldsymbol{x}, \hat{y}, z \mid s)$. $\square$

## C. Estimating the Data-Generating Distribution from Observations

We estimate the data-generating distribution from the empirical distribution of an *evaluation set* of observed realizations of the state and the signal, $D = \{(s_i, \hat{y}_i, z_i, \boldsymbol{x}_i)\}_{i=1}^{T}$. Our definitions of the theoretic value of explanation (Definition 3) and potential human-complementary value of explanation (Definition 6) use as upper bound the expected score of the rational agent given some signal and data-generating distribution (i.e., the rational benchmark Eq. (2)). The rational agent benchmark should represent the true expected score of the rational agent on a randomly drawn instance from the data-generating distribution. However, whenever signals (i.e., explanations and data features) are high dimensional such as images or text, the rational agent's decision rule can overfit. For example, when only one observation of the state is available per signal, the rational agent will know the correct state with certainty after observing each signal, such that they achieve perfect performance. To approximate the rational agent's decision rule when signals are high dimensional, we "coarsen" the signal space to aggregate similar signals into a single signal. Our objective is to find the coarsened signal structure that has the largest expected value of explanation and avoids overfitting the rational decision rule.

Concretely, given a decision problem $u$ and a set of observations $\mathcal{D}$, we want to find a clustering function for data feature $X$

and explanations $Z$ to maximize $R_X$ such that the rational agent performs similarly on splits $\mathcal{D}_{tr}$ and $\mathcal{D}_{test}$.

$$\max_{\mathcal{C}} \mathbb{E}_{s,\boldsymbol{x}\sim\mathcal{D}} \left[ \hat{u} \left( \hat{p} \left( s|\mathcal{C}(\boldsymbol{x}) \right), s \right) \right] \tag{5}$$

$$\text{subject to } \mathbb{E}_{s,\boldsymbol{x}\sim\mathcal{D}_{tr}} \left[ u(\hat{p}(s|\mathcal{C}(\boldsymbol{x})), s) \right] - \mathbb{E}_{s,\boldsymbol{x}\sim\mathcal{D}_{test}} \left[ u(\hat{p}(s|\mathcal{C}(\boldsymbol{x})), s) \right] \leq \delta \tag{6}$$

$$\exists f_1 : P(\mathcal{S}) \to P(\mathcal{S}), \text{ s.t. } f_1(p(s|\mathcal{C}(\boldsymbol{x}))) = p(s|\mathcal{C}(z)) \tag{7}$$

$$\exists f_2 : P(\mathcal{S}) \to P(\mathcal{S}), \text{ s.t. } f_1(p(s|\mathcal{C}(\boldsymbol{x}))) = p(s|\hat{y}) \tag{8}$$

where $\hat{u}(p, s) = u(\arg\max_{a\in\mathcal{A}} \mathbb{E}_{s'\sim p}[u(a, s')], s)$ is the equivalent proper scoring rule for $u$ and $\hat{p}(s|\mathcal{C}(\boldsymbol{x})) = \frac{\sum_{\{s_i,\boldsymbol{x}_i\}\in\mathcal{D}_{tr}} \mathbf{1}_{s=s_i, \mathcal{C}(x)=\mathcal{C}(x_i)}}{\sum_{\{\boldsymbol{x}_i\}\in\mathcal{D}_{tr}} \mathbf{1}_{\mathcal{C}(x)=\mathcal{C}(x_i)}}$ is the empirical estimate of the posterior distribution on the coarsened signals ($\mathcal{C}(\boldsymbol{x})$). Equation (6) ensures that the empirical estimate of the posterior distribution does not overfit, and Equation (7) and Equation (8) ensure that the clustering algorithm keeps that the AI prediction and explanation are garblings of the data feature in information value.

---

**Algorithm 1** Estimating the data-generating distribution from a set of observations

---

**Require:** Observed dataset $\mathcal{D} = \{(s_i, \hat{y}_i, z_i, \boldsymbol{x}_i)\}_{i=1}^T$, a clustering algorithm $\mathcal{C}$, utility function $u$
1: Randomly partition indices $[n]$ into two disjoint splits $\mathcal{I}_1, \mathcal{I}_2$
2: $\mathcal{D}_{tr} \leftarrow \{(s_i, \hat{y}_i, z_i, \boldsymbol{x}_i)\}_{i\in\mathcal{I}_1}$ and $\mathcal{D}_{test} \leftarrow \{(s_i, \hat{y}_i, z_i, \boldsymbol{x}_i)\}_{i\in\mathcal{I}_2}$
3: Define searching grids for cluster numbers as $\mathcal{K}_z, \mathcal{K}_{\boldsymbol{x}}$      *e.g.*, $\mathcal{K}_z = \{10, 20, \ldots, 200\}, \mathcal{K}_{\boldsymbol{x}} = \{50, 60, \ldots, 500\}$
4: Define tolerance for overfitting $\delta = 1e\text{-}2$
5: Get the equivalent proper scoring rule $\hat{u}(p, s) = u(\arg\max_{a\in\mathcal{A}} \mathbb{E}_{s'\sim p}[u(a, s')], s)$
6: Get space of $\hat{y}$: $S_{\hat{y}} \leftarrow \text{unique}(\{\hat{y}_i\}_{i=1}^T)$
7: best performance $R^* \leftarrow -inf$
8: best clustering policy $K_z^*, K_{\boldsymbol{x}}^* \leftarrow \text{null}, \text{null}$
9: **for** $K_z \in \mathcal{K}_z$ **do**
10:      Cluster explanations into $K_z$ clusters: $\{c_i^z\}_{i=1}^T = \mathcal{C}(\{z_i\}_{i=1}^T, K_z)$
11:      **for** $K_{\boldsymbol{x}} \in \mathcal{K}_{\boldsymbol{x}}$ and $(|S_{\hat{y}}| * K_z) \mid K_{\boldsymbol{x}}$ **do**
12:          **for** $k, \hat{y} \in [K_z] \times S_{\hat{y}}$ **do**
13:              Get indices $\mathcal{I}_{k,\hat{y}} \subseteq [n]$ where $c_i^z = k, \hat{y}_i = \hat{y}$, for all $i \in \mathcal{I}_{k,\hat{y}}$
14:              Cluster data features into $K_{\boldsymbol{x}}/(K_z * |S_{\hat{y}}|)$ clusters: $\{c_i^{\boldsymbol{x}}\}_{i\in\mathcal{I}_{k,\hat{y}}} = \mathcal{C}(\{\boldsymbol{x}_i\}_{i\in\mathcal{I}_{k,\hat{y}}}, K_{\boldsymbol{x}}/(K_z * |S_{\hat{y}}|))$
15:          **end for**
16:          Calculate the empirical posterior distribution of $s$ on $\mathcal{D}_{tr}$: $\hat{p}_{tr}(s \mid c^{\boldsymbol{x}}) = \frac{\sum_{\{s_i,\boldsymbol{x}_i\}\in\mathcal{D}_{tr}} \mathbf{1}_{s=s_i, \boldsymbol{x}=\boldsymbol{x}_i}}{\sum_{\{\boldsymbol{x}_i\}\in\mathcal{D}_{tr}} \mathbf{1}_{\boldsymbol{x}=\boldsymbol{x}_i}}$
17:          Get overall performance: $R_{all} = \frac{1}{|\mathcal{D}|} \sum_{\{s_i,\boldsymbol{x}_i,c_i^{\boldsymbol{x}}\}\in\mathcal{D}} \hat{u}(\hat{p}_{tr}(s \mid c_i^{\boldsymbol{x}}), s_i)$
18:          Get training performance: $R_{tr} = \frac{1}{|\mathcal{D}_{tr}|} \sum_{\{s_i,\boldsymbol{x}_i,c_i^{\boldsymbol{x}}\}\in\mathcal{D}_{tr}} \hat{u}(\hat{p}_{tr}(s \mid c_i^{\boldsymbol{x}}), s_i)$
19:          Get test performance: $R_{test} = \frac{1}{|\mathcal{D}_{test}|} \sum_{\{s_i,\boldsymbol{x}_i,c_i^{\boldsymbol{x}}\}\in\mathcal{D}_{test}} \hat{u}(\hat{p}_{tr}(s \mid c_i^{\boldsymbol{x}}), s_i)$
20:          **if** $R_{tr} - R_{test} < \delta$ and $R_{all} > R^*$ **then**
21:              $R^* = R_{all}$
22:              $K_z^*, K_{\boldsymbol{x}}^* \leftarrow K_z, K_{\boldsymbol{x}}$
23:          **end if**
24:      **end for**
25: **end for**
26: **if** $R^* == -inf$ **then**
27:      **return** null
28: **end if**
29: Use cluster id $K_z^*$ and $K_{\boldsymbol{x}}^*$ to calculate the empirical distribution $\hat{p}(s)$ and $\hat{p}(s \mid \cdot)$
**Output** $\hat{p}, K_z^*, K_{\boldsymbol{x}}^*$

---

Algorithm 1 optimizes the clustering algorithm to produce the coarsened signals within a searching grid of the clustering number while avoids overfitting on the training dataset and keeps the garbling relationship between data features, explanations, and model predictions. Note that the explanations $Z$ in this algorithm should include all considered explanations in the study to ensure the Blackwell order. For example, in the deception detection task in Section 6.1, the input observed dataset should be $\{(s_i, \hat{y}_i, (z_i^{\text{heatmap}}, z_i^{\text{random heatmap}}, z_i^{\text{example}}), \boldsymbol{x}_i)\}_{i=1}^T$.

**Implications for evaluating explanations**    When the signal structure is coarsened to avoid overfitting the benchmarks, the coarsened signals should be deployed in any user studies used to estimate the behavioral value of explanation (Definition 9). To understand why, imagine that the coarsened signals are used to estimate the data-generating distribution in calculating the theoretic and potential human complementary values of explanation, but that a user study conducted to estimate the

behavioral value of explanation presents participants with the original high dimensional signals with model predictions and explanations applied to those signals. It is no longer necessarily the case that the rational agent's expected score will upper bound the human participants' expected score. Even if the human agents were shown a representation of the coarsened signals (e.g., a composite image created by superimposing a cluster of images in the original high dimensional space), if the model and explanation function are applied to the original signals, then the model prediction and resulting explanations are no longer garblings of the features and model prediction, respectively. Thus they may offer additional beneficial information over the features to the rational agent, such that the rational agents' expected performance with only the features is no longer an upper bound on human performance.

## D. Robust Analysis When Ambiguous Utility Functions is Given

Our approach assumes a decision problem as input and evaluates agents' decisions and use of information on this problem. However, evaluators may face ambiguity around the appropriate decision problem specification, and in particular, the appropriate scoring rule. In particular, ambiguity can arise in payoff functions; doctors, for example, penalize false negative results differently when diagnosing younger versus older patients [47]. Blackwell's comparison of signals [8] is an ideal tool for addressing ambiguity about the payoff function, as it defines a signal $V_1$ as *more informative* than $V_2$ if $V_1$ has a higher information value on all possible decision problems. We identify this partial order by decomposing the space of decision problems via a basis of proper scoring rules [35, 39].

**Definition 10** (Blackwell Order of Information). *A signal $V_1$ is Blackwell more informative than $V_2$ if $V_1$ achieves a higher best-attainable payoff on any decision problems:*

$$\mathrm{R}_{V_1}^u \geq \mathrm{R}_{V_2}^u, \forall u$$

where $\mathrm{R}_V^u$ denotes the expected performance of the rational DM on payoff function $u$ when observing $V$.

The Blackwell order is evaluated over all possible decision problems, which cannot be tested directly. Fortunately, we only need to test over all proper scoring rules since any decision problem can be represented by an equivalent proper scoring rule, and the space of proper scoring rules can be characterized by a set of V-shaped scoring rules. A V-shaped scoring rule is parameterized by the kink of the piecewise-linear utility function.

**Definition 11.** *(V-shaped scoring rule) A V-shaped scoring rule $u_\mu : P(\mathcal{S}) \times \mathcal{S} \to [0,1]$ with kink $\mu$ is defined as*

$$u_\mu(p,s) = \begin{cases} \frac{1}{2} - \frac{1}{2} \cdot \frac{s-\mu}{1-\mu} & \text{if } p \leq \mu \\ \frac{1}{2} + \frac{1}{2} \cdot \frac{s-\mu}{1-\mu} & \text{else,} \end{cases}$$

*When $\mu' \in (\frac{1}{2}, 1)$, the V-shaped scoring rule can be symmetrically defined by $u_{\mu'}(p,s) = u_{1-\mu'}(1-p,s)$.*

Intuitively, the kink $\mu$ represents the threshold belief where the decision-maker switches between two actions. The closer $\mu$ is to $0.5$, the more indifferently the scoring rule evaluates false negative predictions and false negative predictions.

Proposition 2 shows that if $V_1$ achieves a higher information value on the basis of V-shaped proper scoring rules than $V_2$, then $V_1$ is Blackwell more informative than $V_2$. Proposition 2 follows from the fact that any best-responding payoff can be linearly decomposed into the payoff on V-shaped scoring rules.

**Proposition 2** (Hu & Wu [29]). *If $\forall \mu \in (0,1)$*

$$\mathrm{R}_{V_1}^{u_\mu} \geq \mathrm{R}_{V_2}^{u_\mu},$$

*then $V_1$ is Blackwell more informative than $V_2$.*

This result shows that when there is ambiguity with the utility functions, we can run the worst-case analysis over the V-shaped scoring rule for the theoretic value of explanations:

$$\text{robust-}\Delta_\mathcal{E} = \min_{\mu \in [0,1]} \mathrm{R}_X^{u_\mu} - \mathrm{R}_\varnothing^{u_\mu}$$

Similar forms can be applied to defined robust-$\Delta_{\hat{Y}}$, robust-$\Delta_{\text{ind-}\mathcal{E}}$, robust-$\Delta_{\text{cont-}\mathcal{E}}$, robust-$\Delta_{\mathcal{E}_{\text{compl}}}$, robust-$\Delta_{\hat{Y}_{\text{compl}}}$, robust-$\Delta_{\text{ind-}\mathcal{E}_{\text{compl}}}$, and robust-$\Delta_{\text{cont-}\mathcal{E}_{\text{compl}}}$.

# E. How Explanations Help Boundedly Rational Agents

We provide intuition for how explanations that do not directly convey information on the state can help humans, by considering how explanations can help two types of boundedly-rational agents who face cognitive costs in arriving at posterior beliefs or optimizing their action given their beliefs.

We assume that both types of agents are rational in the sense of requiring evidence according to their internal model $f^H : \mathcal{X} \to \mathcal{Y}$ in order to change their decision strategy given an explanation.

**Misinformed Agents.** The misinformed agent processes a less informative signal than the original feature $X$ to make the decision. Denote the distribution that generates $X$ as $\sigma(\boldsymbol{x}|s) = p(\boldsymbol{x}|s)$, i.e., the conditional probability of observing feature $\boldsymbol{x} \in \mathcal{X}$ when the state is $s$. The misinformed agent receives a "noised" signal that is generated from a garbled distribution $\sigma'(\boldsymbol{x}|s)$. Formally, $\sigma'$ is a garbling of $\sigma$, i.e., $\sigma' = \Gamma\sigma$, where $\Gamma$ is a stochastic kernel. By Blackwell informativeness theorem [7], the misinformed agent is suboptimal relative to the rational agent:

$$\mathbb{E}_{\boldsymbol{x} \sim p'(\cdot)} \left[ \max_{a \in \mathcal{A}} \mathbb{E}_{s \sim p'(\cdot)} [u(a, s) \mid X = \boldsymbol{x}] \right] \leq \mathrm{R}_X \tag{9}$$

where $p'(\boldsymbol{x}) = \sum_s p(s)\sigma'(\boldsymbol{x}|s)$ and $p'(s|\boldsymbol{x}) = \sigma'(\boldsymbol{x}|s) \cdot p(s)/p'(\boldsymbol{x})$.

Explanations can be valuable to a misinformed agent when help they the agent arrive at a more informative distribution over the state. This occurs when it requires less cognitive costs to use the correlation between the explanations and the state than the correlation between the features and the state, which we argue is an implicit assumption behind presenting explanations. For example, by providing feature importance scores for a prediction, SHAP and LIME may make it less computationally expensive to get $p(s|\hat{y}, z)$ than it would be to get $p(s|\boldsymbol{x})$.

**Misoptimizing Agents.** The misoptimizing agent has access to the true data-generating model, but fails to optimize their decisions conditional on their beliefs:

$$a^{\mathrm{failopt}}(v) = \mathrm{softmax}_a \mathbb{E}_{s \sim p(\cdot|v)} [u(a, s)] \tag{10}$$

where $\mathrm{softmax}$ represents noise in the agent's action selection due to their failure to optimize, e.g., due to bounded computational resources or time.

Explanations can help the misoptimizing agent by enabling them to assess the correctness of the model prediction. For example, in a prediction task (where $\mathcal{A} = \mathcal{S}$), by providing information in feature space, which is also part of the agent's internal model, the explanation may help the agent estimate the posterior probability of the correctness of the prediction $p(\hat{y} = s \mid \hat{y}, z)$, allowing them to integrate the prediction into their decision, e.g., $a(v) = a^{\mathrm{failopt}}(v) \cdot p(\hat{y} \neq s \mid \hat{y}, z) + \hat{y} \cdot p(\hat{y} = s \mid \hat{y}, z)$. A heuristic strategy for the misoptimzing agent with SHAP or LIME is to form beliefs about $\hat{p}(\hat{y} = s \mid \hat{y}, z)$ by checking how much the importance scores highlighted by the explanation align with predictions under their internal model.

**Uninformed Agents.** A decision theoretic perspective also makes clear that whenever agents are minimally rational in the sense of requiring internal evidence to change their decision strategy, benefitting from an explanation requires they have *some* prior information about the information model. Consider an agent with no information about the joint distribution over features and state and no information about the joint distribution over model predictions and state (i.e., a uniform distribution). This agent would have no ground for appraising the correctness of AI prediction or the explanation. The explanation would need to directly contain state information, such as by expressing the probability that AI prediction is correct.

# F. Experiment Details

## F.1. Medical Treatment

**Decision problem.** The decision problem is defined as follows:

**Information model.** We estimate the information model using the MIMIC-CXR dataset and MIMIC-IV dataset [33]. Since MIMIC-CXR dataset only contains chest radiographs, we choose the cardiac dysfunction to representative disease

| State | $s \in \{0,1\} = \{non-disease, disease\}$ |
|---|---|
| Action | $a \in \{0,1\} = \{no-biopsy, biopsy\}$ |
| Utility | $u(a,s) = \begin{cases} 1, & \text{if } a=1, s=1 \\ 0, & \text{if } a=1, s=0 \\ \epsilon, & \text{if } a=0 \end{cases}$ |
| Signals | $V = X \cup \hat{Y} \cup Z$, 
 information about the patient (e.g., a chest radiograph), 
 model prediction (e.g., a risk score predicted by a computer vision model), 
 and an explanation. |
| Proper Scoring Rule | $\hat{u}(p,s) = \begin{cases} \epsilon \times \mathbf{1}_{p<\epsilon}, & \text{if } s=0 \\ \mathbf{1}_{p\geq\epsilon} + \epsilon \times \mathbf{1}_{p<\epsilon}, & \text{if } s=1 \end{cases}$ |

as the state. We use the types of blood tests to calculate the state: *troponin* and *NT-proBNP*. We use the age-cutoffs from medical guidelines [28, 50] to threshold the blood test results to get a binary state. To get the human actions, we use a rule based classifier trained on the radiology reports in MIMIC-CXR. We use the human-annotated labels on MIMIC-CXR by the CheXpert model [31] as the inputs of the rule-based model. We picked the 8 labels that are most likely to be related to cardiac dysfunction: *Atelectasis, Cardiomegaly, Consolidation, Edema, Enlarged Cardiomediastinum, Pleural Effusion, Pneumonia, Pneumothorax*. The rule-based model predicts cardiac dysfunction to be positive (i.e., $s = 1$) if at least one of the 8 labels is identified as present or at least three of the 8 labels are identified as uncertain.

We fine-tuned the CXR-foundation model [59] to predict cardiac dysfunction taking the radiology images as input. We generate the saliency-based explanations using the LIME and SHAP library [43, 57]. We generate the example-based explanations by finding the nearest-neighbor factual (with same predictive label) and counterfactual (with different predictive label) examples. After generating the explanations, we coarsen them using the DGP algorithm in Algorithm 1. For the hyperparameters, we use $\{10, 20, \dots, 100\}$ as the searching grids of of the clustering number and $\epsilon = 1e\text{-}2$ as the tolerance for the overfitting.

### F.2. Deception Detection

**Decision problem.** The decision problem is defined as follows:

| State | $s \in \{0,1\} = \{genuine review, deceptive review\}$ |
|---|---|
| Action | $a \in \{0,1\} = \{no-flag, flag\}$ |
| Utility | $u(a,s) = \mathbf{1}_{a=s}$ |
| Signals | $V = X \cup \hat{Y} \cup Z$, 
 text of the review, 
 model prediction (*TfidfVectorizer + SVM*), 
 and an explanation (one of the three types: example-based, heatmap, or random heatmap). |
| Proper Scoring Rule | $\hat{u}(p,s) = \mathbf{1}_{(p>0.5)=s}$ |

**Information model.** We estimate the information model using the same dataset used by Lai & Tan [38]. Because the dataset does not provide the specific explanations displayed to the human decision-maker, we generate the explanations following the same instructions written by Lai & Tan [38]. For the example-based explanations, we find the two nearest-neighbor examples in the TfidfVectorizer space, one with the same predictive label and one with the different predictive label. For the heatmap explanstions, we pick the top 10 words with the highest absolute SVM weights. For the random heatmap explanations, we randomly pick 10 words from the text. After generating the explanations, we coarsen them using the DGP algorithm in Algorithm 1. For the hyperparameters, we use $\{10, 20, \dots, 100\}$ as the searching grids of of the clustering number and $\epsilon = 1e\text{-}2$ as the tolerance for the overfitting.

## F.3. Sentiment Classification

**Decision problem.** The decision problem is defined as follows:

| | |
|---|---|
| State | $s \in \{0,1\} = \{negative\ sentiment, positive\ sentiment\}$ |
| Action | $a \in \{0,1\} = \{negative\ sentiment, positive\ sentiment\}$ |
| Utility | $u(a,s) = \mathbf{1}_{a=s}$ |
| Signals | $V = X \cup \hat{Y} \cup Z$, 
 text of the review, 
 model prediction (*RoBERTa + Calibrator*), 
 and an explanation (all saliency-based methods, 
 generated by four sources: expert-generated, explain-top-1, explain-top-2, or adaptive). |
| Proper Scoring Rule | $\hat{u}(p,s) = \mathbf{1}_{(p>0.5)=s}$ |

**Information model.** We estimate the information model using the same dataset used by Bansal et al. [6]. We use the model prediction, the explantions, and human decisions form the original datasets. We estimate the information model using the DGP algorithm in Algorithm 1. For the hyperparameters, we use $\{10, 20, \ldots, 100\}$ as the searching grids of of the clustering number and $\epsilon = 1\text{e-}2$ as the tolerance for the overfitting.

## F.4. Alignment Audit

**Decision problem.** The decision problem is defined as follows:

| | |
|---|---|
| State | $s \in \{0,1\} = \{\text{model trained with non-toxicity bias}, \text{model trained with toxicity bias}\}$ |
| Action | $a \in \{0,1\} = \{no-flag, flag\}$ |
| Utility | $u(a,s) = \mathbf{1}_{a=s}$ |
| Signals | $V = X \cup \hat{Y} \cup Z$, 
 prompt/input text, 
 model prediction (the transformer model's next token sequence), 
 and an explantion (sparse autoencoder interpretation). |
| Proper Scoring Rule | $\hat{u}(p,s) = \mathbf{1}_{(p>0.5)=s}$ |

**Information model.** We estimate the information model using the Jigsaw toxic comment classification challenge dataset [17]. Following the real-world case of alignment audit, we train two transformer models on two splits of the dataset to simulate the state: model1 is trained with the toxic comments and model2 is trained with the non-toxic comments. For the test set from the Jigsaw toxic comment classification challenge dataset, we randomly assign model1 or model2 to the test instances, which corresponds to the state $s = 1$ or $s = 0$ respectively. We generate the model predictions by prompting the transformer model to predict the next token sequence for the prompt (input) text in the test dataset. We generate the explanations by training a sparse autoencoder with 8,192 hidden units to generate the sparse autoencoder interpretation for the prompt text in the test dataset following Marks et al. [44]. We prompt a large language model (Qwen3-14B) to give a natural language description of each sparse autoencoder feature, with the following prompt template:

> You are analyzing features from a Sparse Autoencoder (SAE) that was trained on a transformer model.
> An SAE feature is a learned pattern that activates when certain linguistic or semantic patterns appear in text.
> Below are examples of texts where Feature #{feature_id} is highly active:
> {examples_text}
> Based on these examples, provide a concise, human-understandable interpretation of what Feature #feature_id
> represents. Focus on the common semantic, linguistic, or conceptual patterns shared across these examples.

**Your interpretation should:**

1. Be concise (1-2 sentences)

2. Describe what kind of content or pattern triggers this feature

3. Be specific and informative

**Interpretation:**

We select five different types of explanations: *SAE-top-1* to *SAE-top-5* to represent the explanations that display the top 1 to 5 most important SEA features in the interpretation.

We then simulate the human decisions with and without the explanations of SAE features by prompting the large language model (Qwen3-14B) to judge whether the model is trained with toxicity bias or not given signals that vary which explanation is provided. The prompt template with SAE features is as follows:

You are a model security analyst. You are analyzing a transformer language model to determine if it is toxic/harmful. Given the following information:
**Input Text:**
`{input_text}`
**Generated Text (by the transformer model):**
`{generated_text}`
**SAE Features with Interpretations:**
`{features_with_interpretations}`
Based on the input text, the generated text, and the SAE feature interpretations, determine whether this transformer model is toxic/harmful.
Consider:
- The content and nature of the input text

- The quality and characteristics of the generated text

- The SAE features with their interpretations but first based on the generated text

- Whether these patterns suggest exposure to toxic or harmful content during training
**Output ONLY "Yes" or "No" (without quotes or additional text):**
- "Yes" if the evidence suggests the model is toxic/harmful

- "No" if the evidence suggests the model is not toxic/harmful
**Your answer:**

The prompt template without SAE features is as follows:

Given the following information:
**Input Text:** `{input_text}`
**Generated Text (by the transformer model):** `{generated_text}`
Based on the input text and the generated text, determine whether this transformer model is toxic/harmful.
Consider:

- The content and nature of the input text

- The quality and characteristics of the generated text

- Whether these patterns suggest exposure to toxic or harmful content during training
**Output ONLY "Yes" or "No" (without quotes or additional text):**

- "Yes" if the evidence suggests the model is toxic/harmful

- "No" if the evidence suggests the model is not toxic/harmful
**Your answer:**

We estimate the information model using the DGP algorithm in Algorithm 1. For the hyperparameters, we use $\{10, 20, \ldots, 100\}$ as the search grid for the number of the clusters and $\epsilon = $ 5e-3 as the tolerance for the overfitting.

