# OpenReview forum: "Explanations are a Means to an End: Decision Theoretic Explanation Evaluation"
_ICML.cc/2026/Conference — ICML 2026 regular_

### Official Review · Reviewer_NmMR · 2026-03-11

**Soundness:** 2
**Presentation:** 3
**Significance:** 2
**Originality:** 3
**Overall Recommendation:** 4
**Confidence:** 4

**Summary:**

The paper suggests a pragmatic framework for deciding on explanation use based on what the explanation would enable for a 'rational agent',  proposing theoretical  values to estimate or predict how useful an explanation might be to a user. These quantities include a theoretical value of an explanation which measures the difference an explanation makes to the score of a rational Bayesian agent, a human-complementary value which uses a different baseline (a human benchmark), and the behavioral value of explanation which estimates the difference in score of a human decision-maker if they have access to the explanaiton or not.

**Compliance With Llm Reviewing Policy:**

Affirmed.

**Final Justification:**

My final recommendation is weak accept. My main concerns were in the soundness and signficance, which the authors addressed in their rebuttal and so I increased my score from weak reject to weak accept. I would increase further to an accept if I could see how they address the concerns in the paper (e.g. with further discussion/justification and a limitations section) but it is not possible to see this yet at this stage.

**Key Questions For Authors:**

Could you include a discussion or argument for why these theoretical scores are appropriate stand-in for user studies, particularly engaging with some of the HCI work in this area?

**Limitations:**

No, I think the authors should add a discussion of what they believe the limitations are. One area they could discuss is the limitaiton of using theoretical scores as stand-ins (as per my question above).

**Strengths And Weaknesses:**

The paper is organized and written clearly. However, I'm not convinced by the 'theroetic value of explanation' as I'm not sure that the value of an explanation to a human can be determined in absence of human/user input or study. I think if the authors want to maintain this position they would need to argue why a rational Bayesian agent is an appropriate stand-in for a human.  I personally would not be confident in relying on the theroretic value of an explanation when deciding if I should use an explanation method or not. If the authors can provide an argument or discussion of this, I would be ok with an accept. I may still not agree with the positioning but at least I would then understand the reasoning more. Because of this I'm also unsure of the significance. The paper would benefit from  a discussion of its limitations.

---

> ### Author Rebuttal · Authors · 2026-03-31
>
> Thanks for your review. We agree, the theoretic value of explanation is not a substitute for observing human decision-makers with the explanation–but the latter is only worth doing if you have defined a decision problem where an explanation could in theory help a person. The use of the rational Bayesian agent is as an upper bound: if even the best possible decision maker could not benefit from being able to extract more information from the features (because the instance level information itself is not that important to making good decisions, i.e., they could do well just by knowing the prior), then there’s no point in studying human decision makers. The study design is “dead in the water”, so to speak. The only way the explanation could be helpful is if the decision makers are so bad without it that they underperform just taking the same action very single time, based on what’s better under the prior. But in that case, the first step to improving decision-making is to help them understand the prior, not add a complicated explanation.
>
> That said, we agree it would be useful to add a limitations section with the additional space provided in the camera ready. We plan to note the following limitations:
> - We do not propose to use the theoretic value of explanation and the human-complementary value of explanation to predict the actual human performance, which should be used as benchmarks to help check whether we can expect the users of explanations to gain performance improvement.
> - The framework uses rational performance with prior belief in the information model as baseline. The actual behavioral value may exceed the theoretical value. However, even in this case, we still see this as a method to better interpret the improvement on behavioral performance between the benefit from better prior or better signal design from explanations.
>
>
> Thanks for engaging with our paper! Your comments help us make it stronger.

---

> > ### Author Rebuttal · Reviewer_NmMR · 2026-04-01
> >
> > Thank you to the authors for their rebuttal, my main concerns are addressed and I'll increase to a weak accept. My remaining question/concern is the possibility of excluding explanations that a user may actually find helpful in case of error of the theoretical value and the human-complementary value. How possible of an occurence do the authors think this is? I think its difficult to measure exactly what might be important to a user because of qualitative context of the task etc. So could the authors maybe also include a brief discussion of this in the limitations?

---

> > > ### Author Response · Authors · 2026-04-07
> > >
> > > We thank the reviewer for further questions. Error in the theoretical value and human complementary value can be misleading if the decision problem is poorly specified, because the evaluator uses the wrong utility function, for example. In this case, doing a robust analysis can help, as we demonstrate in Appendix D.
> > >
> > > They can also fail to upper bound the behavioral value when the users are worse than the rational agent always choosing the best fixed action under the prior. However, in this case, the first thing to do to improve performance would be to better convey to users a reasonable sense of the prior. We can further emphasize this is the paper through a stand-alone paragraph in limitations.

---

### Official Review · Reviewer_FYF5 · 2026-03-12

**Soundness:** 3
**Presentation:** 3
**Significance:** 2
**Originality:** 3
**Overall Recommendation:** 5
**Confidence:** 3

**Summary:**

This paper gives various definitions and benchmarks, along with methods to calculate them, to evaluate the usefulness of a model explanation in terms of its benefit to a human in performing a task. The authors quantify 1) the theoretical value of an an explanation prior to observing human decisions, 2) the complementary value of an explanation beyond what humans are capable of without the explanation, and 3) the causal effect of giving the explanation to decision-makers. Relevant estimands and empirical estimators are proposed for each of these settings, and the authors demonstrate the value of this approach on decision support and mechanistic interpretability tasks with LLMs serving as proxies for humans.

**Compliance With Llm Reviewing Policy:**

Affirmed.

**Final Justification:**

The authors addressed my primary concern, which was that I believed that the various proposed estimands did not appear to be rigorously estimated, and rather evaluated by LLM proxies. The authors have convinced me that their procedure for computing the estimands in their paper is rigorous and comes with guarantees that I did not pick up in my initial review of the paper. See my response to their rebuttal for more details.

**Key Questions For Authors:**

1. How are model inputs and explanations aggregated to make predictions? Are all of these inputs just fed into an LLM that is assumed to be a good multi-modal prediction model, and comparable to a human using the same information, by default?

2. How are all of the relevant benchmarks estimated? How is $R_X$ obtained? How is $R_Z$ obtained? $R_{A^H \bigcup Z}$? $R_{A^H}$? $B$? These quantities were proposed as theoretical statistical quantities, but appear to have been estimated ad-hoc using LLMs without any statistical guarantees.

3. Could researchers using your definitions plausibly estimate the relevant benchmarks, or could they only get lower bounds based on the best model available to them? i.e., is it possible to assess the true expected utility-maximizing actions, or can you only ever get a proxy by training a model on the finite data available to you?

**Limitations:**

yes

**Strengths And Weaknesses:**

## Strengths

1. The paper was organized in a manner that told a developing "story" -- introducing new estimators to either decompose previous estimators or to address missing components of what they measure. I found the presentation and writing style to be very easy to follow and coherent. The related work is also thorough and ties specific results into existing literature effectively.

2. The figures and the presentation of the results are effective in their style, contain uncertainty quantification, and are described convincingly in the text.

3. I do not know the alignment or human benchmarking literature well enough to be confident in this assessment, but based on the paper, the related work, and a brief personal literature search, the definitions and estimands in this paper are novel, interesting, and appear to be useful ways to decouple and measure various aspects of explanations' usefulness to humans in decision making tasks. This paper could plausibly inform studies of the efficacy of explanations during model deployment down the line.

## Weaknesses

1. I think that some additional information about the contents of the experiments could be useful to make the results a little bit more concrete. The results are well presented and the experimental setup is clear, but I think it would be useful to additionally see an example of each kind of explanation for each experiment. This would help the reader judge for themselves whether the explanation might indeed have helped them on the decision task. For example, it would be useful to see an example of the top 3 SAE features that enable near-perfect performance on the alignment task.

2. I found the consistent use of LLMs as a human proxy in experiments to be uncomfortable when making claims about humans. I think that this is an unnecessary use of language that could be easily avoided to avoid creating confusion or false claims about human behavior and performance. For one example of many, on line 424 in section 5.2, instead of using Qwen3-14B "to simulate human decisions", I think a better framing is that the experiment is evaluating the LLM-complementary value of explanations (rather than human-complementary value). Making this distinction clear across the entire paper would, in this reviewer's opinion, significantly improve the clarity of the experimental claims in the paper.

3. The actual computation of the various estimands is de-emphasized in the paper, and confused me. I found it confusing, for example, that in the medical task, the "human actions" were actually the outputs of a rule-based classifier using annotated labels as input. I also do not understand how the signals are used to make predictions -- the signals appear to be a combination of input data, a predicted label, and explanations in the form of heatmaps, saliency maps, and/or text.

## Review Summary

I think that this paper is clear and well-organized, but falls short in the experimental section of convincing this reviewer that the various proposed estimands are indeed correctly estimated and track onto their proposed real-world meaning. This paper's significance rests on the ability to effectively estimate the estimands on real problems. It seems to me as if all of the estimands are estimated via an LLM proxy that is inspired by the same idea as the estimand, but does not actually have any guarantee of achieving or boundedly-approximating the idealized rational-agent benchmark or complementary value.

Moreover, the results about human behavior and human actions are mostly derived from LLMs "simulating human behavior", which I find dubious at best. I do not believe that a doctor making decisions would use explanations in the same way and achieve the same benefits that the CXR-foundation model would, for example.

---

> ### Author Rebuttal · Authors · 2026-03-31
>
> Thank you for your review, for recognizing the novelty and usefulness of our framework, and for your questions.
>
> Before responding to the weaknesses and questions, we want to clarify a likely misinterpretation: our paper does NOT rely on LLM proxies throughout. The estimands do not depend on LLMs, and we do not advocate using LLMs or other AI models in place of humans. The only place we use LLMs as decision-makers is the third mechanistic interpretability demonstration, and we agree we should not describe them as human simulators. We will revise this framing and rename human complementary value as agent complementary value.
>
> The only other place where we use a model in place of human decisions is the running medical diagnosis example. There, we have radiologists’ reports but not their final diagnoses, so for demonstration purposes we convert reports into decisions using a rule-based model. We will clarify this conversion step in the main text.
>
> Weaknesses:
>
> We like the suggestion to add figures of the decision task with explanations for the experiments. The extra camera-ready page should give us room to do this.
>
> We also agree that the third demonstration should not imply LLMs are human simulations. Accordingly, we can redefine human complementary information as “agent complementary information” and clarify that this may refer to humans or other AI models. LLMs are used as decision-makers only in the mechanistic interpretability demo; in the other demonstrations, we use human decisions from Lai and Tan and Bansal et al.
>
> We also agree that the medical example uses model-converted reports rather than directly observed final human decisions. As described above, we will make this explicit in the main text.
>
> Key Questions:
>
> 1. In response to the question about how model inputs and explanations are aggregated to make predictions, we definitely do NOT just feed everything into an LLM. As noted above, the only place we use an LLM as a decision-maker is the third demonstration. To go from signals to decisions for a rational agent defining the benchmark, the process is as follows:
>
> a. Define the information model, i.e., the joint distribution over the signals and the state. For a rational agent, this is the joint distribution over the features X and the state, because the AI model and explanations are both garblings of the features from an information-theoretic perspective.
>
> b. Apply the Appendix C algorithm to find the optimal coarsening of the information model. If the signals are low-dimensional, the algorithm returns the original information model. If the signals are very high-dimensional (e.g., images or text), a rational agent could memorize the state and produce an overfit upper bound. Algorithm 1 therefore uses hold-out testing to find the optimal signal aggregation and avoid overfitting.
>
> c. The resulting information model gives the posterior probability of the state for any signal, and the rational agent takes the action that maximizes utility under those posterior beliefs.
>
> 2. Again, we do NOT use LLMs to calculate the benchmarks. These are computed using the Appendix C procedure above. We suspect the confusion comes from not describing this process when first introducing theoretic value of explanation and agent complementary value of explanation. We will therefore clarify in revision that we use a systematic method to determine whether coarsening is needed before calculating posterior beliefs and taking the optimal action. The only difference in how we compute R_X, R_Z, R_{A_H U Z}, R_{A_H} is the signal space; otherwise the procedure is the same.
> The coarsening algorithm provides guarantees against overfitting by ensuring the Bayesian rational agent’s performance on the training and hold-out data remains within a $\delta$ interval, while selecting the clustering number that maximizes rational performance. We refer technical details to Appendix C.
> The behavioral value of explanation B is different because no rational agent is used in its calculation. Instead, as described in Section 4.2, one either computes it directly from a controlled study or first fits a structural model to the observed human decisions and then uses that model to isolate the behavioral effect.
>
> 3. It is possible to recover the true expected-utility-maximizing actions whenever no intermediate coarsening is needed. When coarsening is necessary, the benchmarks depend on the coarsening algorithm, but this can still be done systematically with guarantees.
> A further use of the framework, which we do not discuss much due to space, is to infer possible cognitive constraints by defining rational-agent benchmarks with different degrees of coarsening and comparing human decision-makers with explanations to them. This can help indicate how much information people extract, while still distinguishing between failures to extract information and failures to choose the optimal action given that information.
>
> Thanks again for engaging with our paper.

---

> > ### Author Rebuttal · Reviewer_FYF5 · 2026-04-01
> >
> > I thank the authors for their thorough response.
> >
> > It appears as if I did indeed misinterpret the estimation methods for the various estimands -- I read Appendix C, but misunderstood its purpose. The language, "Estimating the data-generating distribution from a set of observations" in the title of the Algorithm as well as the surrounding text implied to me that the goal was to estimate the data-generating distribution itself, rather than to get a good estimate of a model's performance under constraints. I misunderstood this, and upon re-reading Appendix C it is more clear to me what the purpose of that section was and how it corresponds to calculating the proposed estimands. I thank the authors for helping me understand this.
> >
> > The procedure detailed in the response to go from signals to decisions is clear in the rebuttal, but was not clear to me in my reading of the paper. I would encourage the authors to revise section 3.1 to state this procedure in simpler terms, as in the rebuttal. A simple statement of the procedure, followed by justification, would be easier to follow.
> >
> > The authors have addressed my concerns regarding the use of LLM proxies to determine the estimands (which I am now convinced they do not do). I think that the reframing towards "agents" rather than "humans" in the relevant experiments will also improve the clarity of the paper. I encourage the authors to consider ways in which they could bring the procedure for calculating the various proposed estimands a little bit more "into the light" in the paper -- right now, it is sort of hidden in one paragraph in Section 3.1.
> >
> > I thank the authors for engaging and helping me to understand the paper, and I feel that I can confidently recommend acceptance.

---

### Official Review · Reviewer_1rMJ · 2026-03-13

**Soundness:** 3
**Presentation:** 2
**Significance:** 2
**Originality:** 3
**Overall Recommendation:** 4
**Confidence:** 4

**Summary:**

This paper proposes a framework that quantifies the information values of input features, explanations, as well as human-complementary values and other estimands. The authors use several tasks including human-AI deception detection, sentiment classification, and alignment audit using SAE to demonstrate their framework and the relationship between these estimands. Several interesting findings are revealed, such as that explanations can improve participants' decision performance (they offer human-complementary value), but not for sentiment classification tasks. These analyses show the potential of their proposed framework as theoretical guidance for explanation evaluation.

**Compliance With Llm Reviewing Policy:**

Affirmed.

**Key Questions For Authors:**

Using LLMs to simulate humans in the alignment audit application is a debatable choice. In this case, it should be formulated as using-LLM-as-judge, and the LLM-complementary values should be estimated.

**Limitations:**

Please refer to the weaknesses.

**Strengths And Weaknesses:**

**Strengths**:

1. The authors propose a novel theoretical framework that quantifies the information values in input features, explanations, and human knowledge. This gives guidance for potential explanation evaluation. The proofs are provided.

2. The authors use three tasks to demonstrate how their theory can be used to provide insights into explanations in providing information in different (human-AI) decision-making tasks.

**Weaknesses**:

1. Some definitions or motivations/rationales behind the components in the theory need to be further clarified.
- a. The motivation of the theory is: "Explanations are valuable insofar as they improve an agent's expected decision performance." The agent is not well-defined. (This term is now widely used in other AI application contexts.) For instance, a "rational agent" can be understood as a model that obtains all knowledge and reasoning of the model to be explained, since it can correctly derive $y$, $z$ based on only $x$. So can we understand this as the model to be explained? Moreover, in this case, explanations are only valuable for humans, since the ideal agent does not need explanations and can already achieve the expected performance with only $x$.
- b. For $R_z$, if not giving $x$ and $y$, is this plausible in practice? Explanations without input or prediction are not informative, and in practice it is not reasonable to consider this case. For global explanations, it might be possible. However, for counterfactual explanations or other local explanations, it is hard to not include the information of $x$ and $y$.
- c. The definition of $\Delta_{\epsilon}$ is not conditioned on the model itself. However, explanations should be conditioned on models, i.e., $y$.

2. For this theory, the paper only provides how to implement these values. However, it would be great to have real-world user study results to verify the theoretical results. This can make the contributions of the theory more significant.

3. Following up on the last weakness, the theoretical values can be hard to use in practice without verification, as the values do not guarantee the reality. For instance, in Table 1, the random heatmap is not much worse than the examples.

---

> ### Author Rebuttal · Authors · 2026-03-31
>
> Thanks for your review and questions. Here are our responses:
>
> **1a**. The reviewer quotes our statement that "Explanations are valuable insofar as they improve an agent's expected decision performance," but argues that the agent is not well-defined. We disagree! Our work defines two types of agents that matter for evaluating explanations. The first is the idealized agent that upper bounds the performance we can expect from any other agent (including any human). This is the rational agent who starts with the prior (i.e., the probability distribution over the state as defined by the information model, Bayesian updates their beliefs upon viewing the signal, then chooses the utility maximizing action under the utility function.
> This is the value of defining our framework using statistical decision theory–there is an established tradition of using Bayesian belief updating and utility maximization, both of which are well understood mathematical traditions.
>
> In response to the reviewer’s question, the rational agent is not the model to be explained. The rational agent is the best case model of any user of the information that we typically provide in explainability research: the instance information, the model prediction, and the explanation(s). The model to be explained is the prediction/AI model, as is conventionally assumed when talking about research on explanations.
>
> The other agent our work defines is the behavioral agent: the decision-maker who may be biased in various ways relative to the upper bound, including humans. The point of our framework is to provide a way to bridge between the best case performance we could expect from any agent using the explanation and what we observe in practice with humans.
>
> We will add an explicit statement after the sentence the reviewer quoted saying that our work is concerned with enabling comparison between two types of agents: the idealized Bayesian rational agent and the behavioral (e.g., human) agent. We will also include a footnote that this concept of agent refers to a decision agent following the traditions in decision theory.
>
> **1b**. The reviewer is correct to note that not giving x and y is not plausible in practice (except for global explanations that may appear in certain use cases like mechanistic interpretability); we agree that this makes $R_Z$ questionable to use as a main estimand for evaluating explanations. This is exactly why we instead define the theoretical value of explanation as the best case performance one could ever see from x, y, and z. As we describe in the paper, this is in contrast to some prior work, like Chen et al. (2022),  which argues for comparing explanations by their informativeness in isolation. As we describe in Section 3, bottom of page 4, the reason for still calculating $R_Z$ when evaluating explanations is simply to build intuition about the extent to which different explanation methods directly convey information about the state versus can only improve performance by “unlocking” the information in the features (for example, by providing a way for the user to compare what the model is doing to their own internal model of how informative different features are of the state). It’s an open question how valuable it is to have an explanation that directly conveys state information (high $R_Z$); some authors, like Fok and Weld (2024), argue that explanations must directly convey information about the state. Our decomposition of the theoretic value of explanation to components that include $R_Z$ provides a way to test such hypotheses, as we describe in Section 6 Related work.
>
> **1c**. The reviewer is correct, the theoretic value of explanation $\Delta_{\mathcal{E}}$ is not conditioned on the model itself, because to a rational agent who knows the joint probability distribution over the features (X) and the state (Y), a model that takes the form $\hat{f}: X_{AI} \rightarrow Y$, where $X_{AI}$ is a subset of the full set of known features X, provides no additional information.
>
> In writing the paper, we debated whether we should call the theoretic value of explanation $\Delta_{\mathcal{E}}$ something like “Decision Problem Potential,” since it is not a function of the explanation directly. Ultimately we opted against that, because $\Delta_{\mathcal{E}}$ is the upper bound on how well any behavioral could do when you give them an explanation on top of the features and model prediction, which is typically what we care about in explainability research. However, if the reviewer feels strongly that the definition should be called something else (like Decision Problem Potential) we are open to considering that.
>
> Thanks for engaging with our paper! Your comments help us make it stronger.

---

> > ### Author Rebuttal · Reviewer_1rMJ · 2026-04-03
> >
> > I thank the authors for their reply. I would recommend that the authors clarify the definition of "agents"/"rational agent" more explicitly W1a in the paper to avoid misunderstandings for readers.

---

> > > ### Author Response · Authors · 2026-04-07
> > >
> > > We thank the reviewer’s suggestion, and we will add an explicit clarification upon our first mention of agents that our work instantiates the framework by considering two agents: rational and behavioral. We will also define these two types of agents on first mention.

---

### Official Review · Reviewer_SB77 · 2026-03-13

**Soundness:** 4
**Presentation:** 3
**Significance:** 4
**Originality:** 4
**Overall Recommendation:** 5
**Confidence:** 3

**Summary:**

This paper proposes a decision-theoretic framework that formalizes three distinct values for explanation evaluation: theoretical value, human-complementary value, and behavioral value. Beyond formalizing these three values, the authors provide decision-theoretic upper bounds for explanation value, empirical estimators, and a validation framework. They demonstrate their approach through applications to human-AI decision support and mechanistic interpretability. This work represents the first comprehensive formalization of explanation evaluation spanning from theoretical to behavioral explanation value.

**Compliance With Llm Reviewing Policy:**

Affirmed.

**Final Justification:**

The authors addressed all raised concerns. With the minor changes proposed by the authors, the paper would especially be improved in its presentation. I particularly like their idea of adding a standalone section which makes their recommended steps and decision rules explicit (G). Furthermore, the authors answered my questions regarding the interpretation of the three effects and the challenge to the faithfulness statement in a detailed response and practical explanation. This example has helped me better understand the practical value of this theoretical concept. Thank you for clarifying what your idea behind the faithfulness statement is. Adding this discussion to the main document would be really valuable and would enrich the implications of this work, especially given that your paper might not only be relevant for the technical community of ICML but also for the socio-technical discourse.

The clarifications of the authors' rebuttal helped me to better understand their work and implications. Therefore, I stick with my vote, which has been further reinforced.

**Key Questions For Authors:**

1. **Interpretation of the three different effects**: The interpretation of the three different effects (theoretical, human-complementary, behavioral) comes up short. How can we interpret the theoretic and the human-complementary value? Which information does one value provide for the other one? Could the values before work as a proxy if we are not able to measure the behavioral effects?

2. **Challenge to faithfulness statement**: "[...] properties of explanations like faithfulness [...] which are neither necessary nor sufficient for explanations to improve human understanding [...]" (p. 8). This is a strong statement that I would like to challenge. I assume that explanation properties help to understand the explanations by providing information that helps to judge if the explanation is trustworthy or not. Could you clarify your position on this?

**Limitations:**

yes

**Strengths And Weaknesses:**

### Strengths

- A) **The paper provide a novel way of formalizing the different levels of explanation value.** Therefore they introduce a new theoretic framework that is embedded well into prior work.
- B) **The presentation of the paper is really good (besides of some minor things).** The paper is concise and clearly written. Especially the figures and the illustrative examples are really helpful to understand the theoretic framework. Some minor changes would help to improve the readability of the concrete examples and the demonstrations. Moreover, the paper is well structured and easy to follow. Additional information are appropriately given in the appendix and I have the feeling that all necessary information are part of the main paper.
- C) **The theoretic framework proposed by the paper is highly significant.** The paper provides an important contribution towards the discussion around objective, theoretical evaluation of explanations and the added practical value that explanations provide in human behavior.
- D) The paper seems to be technically correct and I did not find any issues. I also have to admit that this is not my field of expertise and I am not able to judge if the mathematical proofs are correct or if all formalization’s are correct from a mathematical standpoint.

### Weaknesses

The paper is generally well written and I do not have major remarks. The following comments can be seen as minor remarks that should help to especially improve the readability of the paper.

- E) **Inconsistent presentation of variables in demonstrations**: I had difficulties reading the text of the demonstrations and the examples, as the delta values are written in the text but are not highlighted in the figure, and the variables highlighted in the figure are not part of the text. It would help to highlight relevant deltas like $\Delta_{\mathcal{E}_{compl}}$ in the figure (e.g., Example on page 6) and/or use important variables such as $R_{A^H} \simeq 0.62$ in the text as well. By adding this information, statements like "For sentiment classification, explanations vary in independent theoretical value (e.g., [...]), but offer little [...]" (p. 7) would become much more readable and understandable.
- F) **Inconsistent x-axis labeling across figures**: It is quite confusing that the x-axes of all figures are labeled differently, while I understood them to be the same (Figure 1: "Expected Decision Performance", Examples: "Expected Payoffs", Figure 2: "Accuracy"). In the latter case, the accuracy is referenced as "participants' accuracy," which feels unusual as I would expect accuracy as a binary-classification metric. For this concrete example, I suggest either choosing a more general label as in the previous figures or arguing why a specific metric is used to avoid confusion.
- G) **Lack of actionable guidelines**: "[...] properties of explanations like faithfulness [...] which are neither necessary nor sufficient for explanations to improve human understanding [...]" (p. 8). To make the workflow "[...] we contribute [...]" more actionable, I am missing a clearer guideline on how to interpret the concrete results of each step and a discussion of the implications.
- H) **Unclear connection in Section 4.2**: "Our decomposition of $\Delta_{\mathcal{E}}$ enables testing when 'verifiability' correlates with effectiveness (Section 4.2)" (p. 8). This did not become clear to me in Section 4.2. I suggest making this statement more explicit in Section 4.2 and also showing how this effect becomes visible in the examples.
- I) **Missing abbreviation definition**: The abbreviation "VoE" (see Figure 1) is not introduced. I assume it stands for Value of Explanation; I suggest introducing this abbreviation explicitly.

---

> ### Author Rebuttal · Authors · 2026-03-31
>
> We are grateful to the reviewer for their comments and for acknowledging our contributions. We agree that the discussion of the variables in the text could be better linked to the quantities in figures (E), and will revise to synchronize figures and text. The x-axis inconsistency (F) is a good catch that we will address, along with defining VoE (the reviewer’s guess is correct).
>
> **On actionable guidelines (G)**: In writing the paper we tried to balance rigorously motivating and defining the different levels of value of explanation with elaborating the workflow that puts these definitions to use and providing recommendations. Some of our advice is easy to miss; currently, advice on workflow (how/when to use each definition) appears at the beginning or end of each of the three definition sections, with advice on the associated decision rules sprinkled throughout. However, we can use part of the additional page in the camera ready to add a stand-alone section on applying the definitions in practice, where we summarize the recommended steps and decision rules in one place.
>
> **On the unclear connection in 4.2 (H)**: We will add an explicit reference in 4.2 to the correlation between $\Delta_{ind-\mathcal{E}}$ (the rational payoff gain of having the explanation) and the behavioral effect $\Delta_{\mathcal{E}_{behavioral}}$. We will add this comparison to our analysis of the examples.
>
> **Interpretation of the three effects**: The theoretical value describes how much potential there is for an explanation to improve task performance. We could observe a larger effect from humans using the explanation only if human performance without the explanation is worse than the best fixed decision you could make if you knew the prior (i.e., a “dumb” baseline). If they are doing worse than the prior, this suggests that the first thing to try is to give them an estimate of the prior, not introduce an explanation. So we recommend never evaluating explanations on decision problems with negligible theoretical value of explanation: in both research and practice we learn little by studying explanations in problems with low theoretical VoE.
>
> The human complementary value describes how much potential there is for an explanation to improve task performance above what we’d expect from humans without the AI. If humans without AI are already close to the best possible performance on the task, then there is little reason to expect an explanation to have much effect.
>
> The behavioral value of explanation is the estimated average treatment effect of the explanation from a study that compares people’s decisions with and without the explanation. We assume both groups get the instance representation and the AI prediction, as is typical in human-AI decision tasks.
>
> In practice, one should not deploy or further study an explanation for a decision problem where it could not provide much value in theory. They should also not deploy or further study the explanation if there is little human complementary value—in practice, this suggests gathering a sample of independent human decisions on the task and then evaluating the human complementary value before deploying the AI. Only when these two checks “pass” should one go on to study how well people use the explanation.
>
> When human decisions with explanations are not possible to study, then yes, the theoretic and human complementary value can be used as proxies, but these are best-case values. Just because there is a large theoretic or human complementary value relative to the baseline (rational agent with only the prior) does not mean that we will see nearly as much benefit in practice. Evaluators just need to remain aware that they are evaluating potential if they use this approach.
>
> **Challenge to faithfulness statement**: We understand why that sentence would seem contestable. We intended it in the mathematical sense of necessary and sufficient, but we agree it should be rephrased to make the intent clearer, which is just to say that faithfulness alone does not guarantee better decisions. The implications of a decision-theoretic view of explanations as a means of supporting appropriate trust are described in the Appendix in the section on misoptimizing agents. The idea is that explanations can help an agent who has some information about the true joint distribution between the features and the state, but does not use it optimally in making a decision. Faithful explanations can enable such agents to assess how much to trust model prediction by comparing their own internal model of how signals correlate with the state with how the model uses the same information. If they have high confidence in their knowledge but it conflicts with the model, this should decrease trust; if it agrees with the model, it should increase trust. In the revision, we plan to use the additional page in the camera ready to discuss this in the main text.
>
> Thanks for engaging with our paper! Your comments help us make it stronger.

---

> > ### Author Rebuttal · Reviewer_SB77 · 2026-04-04
> >
> > I thank the authors for the clarifications. It strengthens my vote.

---

### Decision · Program_Chairs · 2026-04-30

**Decision:**

Accept (regular)

**Comment:**

This paper proposes a decision-theoretic framework for evaluating explanations through three distinct notions of value: theoretical value, complementary value, and behavioral value. It also develops associated estimates, upper bounds, and a practical workflow. Finally, it demonstrates the framework on decision-support and mechanistic-interpretability examples.

Reviewers were overall positive. The main contribution was widely seen as novel, well formulated, and potentially important for explanation evaluation, especially because it helps disentangle several notions that are often conflated in prior work. The paper was also praised for its clarity, structure, and effective examples. The rebuttal appears to have resolved most remaining concerns, especially regarding the interpretation of the different values, the role of rational versus behavioral agents, and the practical workflow for applying the framework.

In definitive, the paper provides a useful conceptual and methodological foundation for future work on explanation evaluation, and I expect it to be valuable to both the technical and broader socio-technical communities working on interpretability.